# MotionSight: Boosting Fine-Grained Motion Understanding in Multimodal LLMs

**Yipeng Du**[1*]  **Tiehan Fan**[1*]  **Kepan Nan**[1,2]  **Rui Xie**[1,2]  **Penghao Zhou**[2]

**Xiang Li**[3]  **Jian Yang**[1]  **Zhenheng Yang**[2]  **Ying Tai**[1✉]

[1] Nanjing University  [2] ByteDance  [3] Nankai University

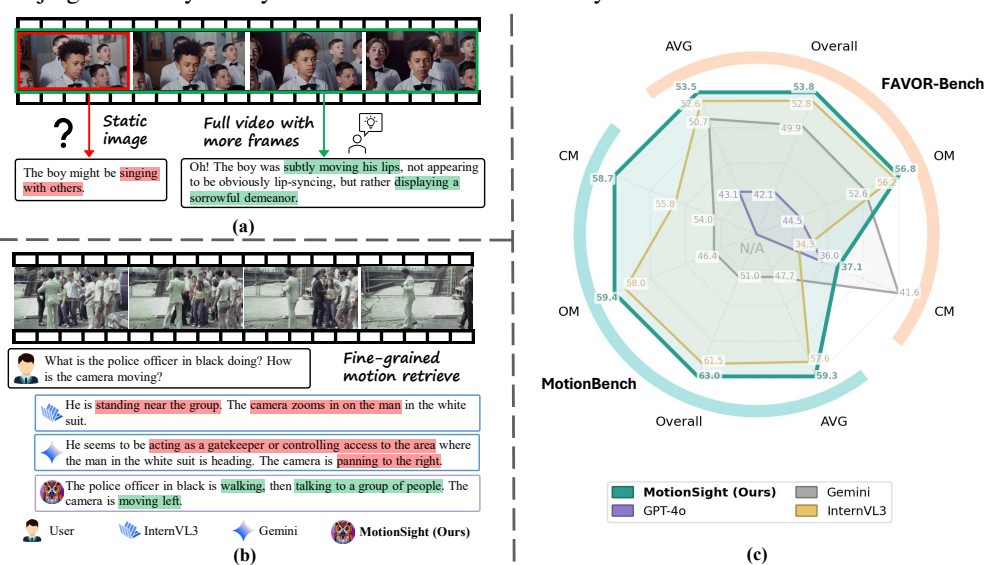

Figure 1: **Motivation and approach overview.** (a) Temporal dynamics inherent in motion distinguish videos from static images. (b) Existing MLLMs show limitations in fine-grained motion detection, whereas our approach excels in accurately understanding object and camera motion. (c) Our approach shows superior performance on MotionBench and FAVOR-Bench compared to SOTA.

## Abstract

Despite advancements in Multimodal Large Language Models (MLLMs), their proficiency in fine-grained video motion understanding remains critically limited. They often lack inter-frame differencing and tend to average or ignore subtle visual cues. Furthermore, while visual prompting has shown potential in static images, its application to videos' temporal complexities, particularly for fine-grained motion understanding, remains largely unexplored. We investigate whether inherent capability can be unlocked to boost MLLMs' motion perception and enable distinct visual signatures tailored to decouple object and camera motion cues. In this study, we introduce `MotionSight`, a novel zero-shot method pioneering object-centric visual spotlight and motion blur as visual prompts to effectively improve fine-grained motion understanding without training. To convert this into valuable data assets, we curated `MotionVid − QA`, the first large-scale dataset for fine-grained video motion understanding, with hierarchical annotations including SFT and preference data, $\Theta(40K)$ video clips and $\Theta(87K)$ QAs. Experiments show `MotionSight` achieves state-of-the-art open-source performance and competitiveness with commercial models. Using `MotionVid − QA`, we fine-tuned `MotionChat` on Qwen2.5VL-7B, which attains 48.3% overall accuracy on FAVOR-Bench that is comparable to Qwen2.5VL-72B's 48.1%. In summary, we present a novel zero-shot method and a large-scale, high-quality dataset specifically for fine-grained motion understanding. All the code and annotations are available at https://nju-pcalab.github.io/projects/MotionSight.

## 1 Introduction

Benefiting from high-quality video-text datasets (Chen et al., 2024b; Liu et al., 2024a; Chen et al., 2024a; Wang et al., 2023; Yang et al., 2024) and large model parameters (Zhu et al., 2025b; Hurst et al., 2024; Bai et al., 2025; Zhu et al., 2025a; DeepMind, 2025), Multimodal Large Language

---

* Equal contributions. Ying Tai is the corresponding author.

Models (MLLMs) have exhibited impressive performance on a wide range of video understanding tasks. Unlike static images, videos possess a temporal dimension characterized by continuous frame-to-frame changes over time. These changes, originating from object or camera motion, create dynamic and expressive motion patterns, distinguishing videos from images, as illustrated in Figure 1(a).

However, even with rapid progress and development in video understanding, the task of fine-grained video motion understanding still lacks necessary attention and exploration. While MLLMs acquire broad semantic knowledge from large-scale data pre-training (Bai et al., 2025; Chen et al., 2024d; Zhang et al., 2024b; Wang et al., 2025a; Zhang et al., 2025), their direct application to motion understanding is often suboptimal. This stems from their tendency to process spatial regions with uniform importance and a lack of inherent mechanisms to explicitly discern the subtle inter-frame variations critical for nuanced motion analysis. These predispositions mean their potential for fine-grained understanding remains largely untapped (Tu et al., 2025) as shown in Figure 1 (b). Given this, how can we boost the latent capabilities of MLLMs derived from large-scale data to achieve fine-grained understanding of local motion cues and enhance the modeling of subtle inter-frame dynamics? Furthermore, how can we transform the augmented implicit understanding from the models into structured data assets that can be used for training other models and for in-depth analysis?

Motivated by these questions, we conducted extensive experiments and explorations on how to boost MLLMs' inherent fine-grained motion understanding capability through zero-shot approaches, without relying on additional training data. Previous studies in image understanding (Yao et al., 2022; Shtedritski et al., 2023; Yang et al., 2023b;a; Yu et al., 2024) have demonstrated strong interest in visual prompting methods, but their extension to address the intricate temporal complexities of video, especially for nuanced fine-grained motion understanding, still requires further investigation. To demonstrate the inadequacy of naively adapting prompting methods from static images to encapsulate the intrinsic temporal dynamics of events in video, we conducted transfer evaluations on recent motion-specific benchmarks, and found that even background blur, the best-performing image-based visual prompt (Yang et al., 2023b), paradoxically exhibited the poorest performance in fine-grained motion understanding. This approach tends to decrease the model's ability in fine-grained motion understanding due to the loss of contextual information, as shown in Figure 3.

To address this, we propose `MotionSight`, **a novel video visual prompting method that decouples object and camera motion tailored for fine-grained motion understanding**. For object motion, we apply a spotlight-like visual prompt on bounding boxes correlated with the queried motion to enhance motion perception (aiming at focusing the model's attention on the core motion, as illustrated in Figure 7). This is inspired by the fact that pre-training data inevitably includes scenarios like stage performances and TV shows, where important moving subjects are often highlighted while the background is dimmed. For camera motion, which necessitates the MLLM's perception of subtle scene changes—a capability where MLLMs often exhibit limitations (Hong et al., 2025; Tu et al., 2025)—we designed a method to synthetically introduce motion blur into video frames. Interestingly, our experiments reveal that this addition of motion blur significantly benefits camera motion determination. Through our carefully designed configuration, `MotionSight` enables MLLMs to achieve enhanced results for fine-grained motion understanding without additional training data.

To further convert this capability into explicit, actionable data assets, we collected and annotated $\Theta(40K)$ video clips, $\Theta(87K)$ question-answer pairs. Through a rigorous filtering mechanism that enhanced the quality of annotated data, we developed an SFT dataset and a preference dataset for training strategies. This process distilled the fine-grained motion understanding capability of MLLMs and aligned it with human preference, and **constructed** `MotionVid − QA`**, the first large-scale open-source dataset for fine-grained video motion understanding to date**, encompassing diverse scenes and high-quality video footage. Benefiting from this dataset, our trained `MotionChat` significantly outperformed the original Qwen2.5VL-7B model, thus validating the effectiveness of `MotionVid − QA`. Extensive experiments demonstrated that `MotionSight` and `MotionVid − QA` effectively contribute to enhancing fine-grained motion understanding.

## 2 RELATED WORK

**MLLMs for video understanding.**   As MLLMs continue to advance, a growing body of research focuses on applying them to video understanding (Bai et al., 2025; Hong et al., 2024; Yao et al., 2024; Xu et al., 2024; Liu et al., 2024d; Zhang et al., 2024a; Wang et al., 2024a; Zhang et al., 2024b; Li et al., 2024b; Wang et al., 2024c; 2025a; Zhang et al., 2025; Wang et al., 2024b; Chen et al., 2024d).

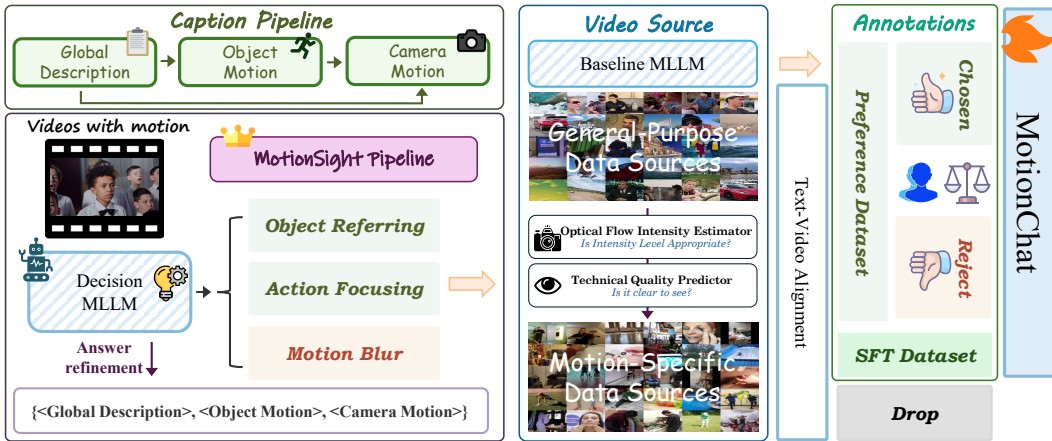

Figure 2: **Overview of the interaction process. Left:** Our `MotionSight` pipeline captions high-quality data, transforming it into data assets. **Right:** This data undergoes rigorous filtering to align with human preferences, resulting in our high-quality dataset `MotionVid − QA`.

Video understanding models often use keyframes as samples, which are then encoded for LLMs. Several approaches develop specialized connectors, while models encode videos frame-by-frame through a vision encoder before feeding them to the LLM. Although these methods excel at event-level video representation, they tend to struggle with fine-grained motion understanding due to limited perception of inter-frame dynamic differences. Directly migrating existing image-level visual prompts to video understanding (Yao et al., 2022; Shtedritski et al., 2023; Yang et al., 2023b;a; Yu et al., 2024) performs poorly due to their failure to account for motion information. To overcome this limitation, we propose to decouple object motion and camera motion, leveraging novel visual prompting methods to improve the model's understanding of fine-grained motion.

**Fine-grained motion understanding datasets.** Early action recognition datasets (Soomro et al., 2012a; Kuehne et al., 2011; Caba Heilbron et al., 2015) had limited fine-grained motion understanding due to simplistic categorical labels. Recent works use MLLMs for auto-annotation (Chen et al., 2024b; Wang et al., 2023; Yang et al., 2024; Nan et al., 2024), but granularity remains limited. Structured video captions (Ju et al., 2024; Fan et al., 2024a; Wu et al., 2025) respond to the need for fine-grained semantics. However, deficiencies persist in motion semantics delineation due to the lack of a well-designed approach for obtaining fine-grained semantic representations. Benchmarks like MotionBench (Hong et al., 2025) and FAVOR-Bench (Tu et al., 2025) have datasets

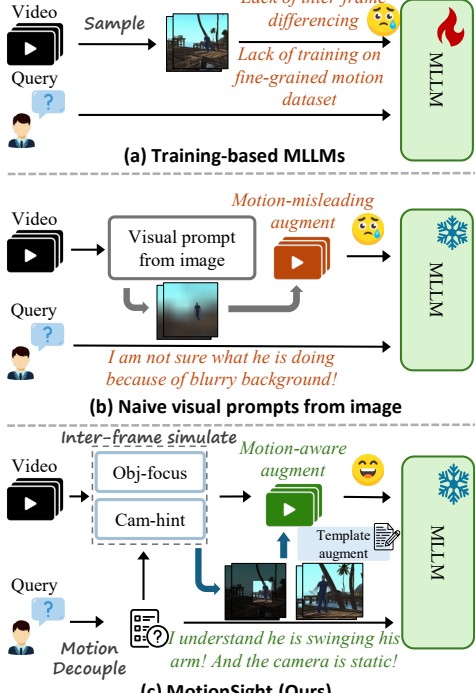

Figure 3: **Comparison of our method with other existing methods.** Directly applying image visual prompts can lead to misinterpretation. By employing decoupled object-guided motion focusing and inter-frame information enhancement, our method addresses the challenge faced by previous methods.

with insufficient sample sizes, limiting scene diversity and semantic richness. To overcome these limitations, we propose `MotionVid − QA`, the first large-scale dataset for fine-grained motion understanding, featuring extensive scene coverage and high video quality.

## 3 MOTIONSIGHT

This section introduces `MotionSight` (Figure 4) with enhanced fine-grained motion perception. Our method decouples object and camera motion and discusses techniques to enhance MLLM input

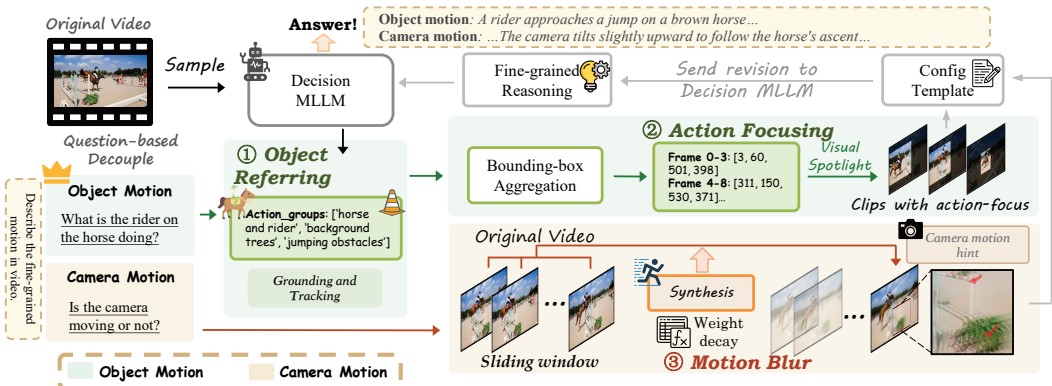

Figure 4: **The detailed pipeline of** `MotionSight`**.** Our method includes query-based *motion decoupling*, gating based on object motion and camera motion. Subsequently, it selectively passes through modules based on the decoupled type. Then, we carefully designed a template prompt for MLLMs to understand our enhanced input and make final decisions.

videos. We sample the input video $\mathbf{V} = \{\mathcal{I}_i\}_{i=1}^{L}$ ($\mathcal{I}_i \in \mathbb{R}^{3 \times H \times W}$) to $\mathbf{V}_s = \{\mathcal{I}_{s_j}\}_{j=1}^{T}$, with $L$ and $T$ denoting original video length and sampled frames length, respectively. Our approach is as follows:

$$\mathcal{R}_{obj} = \text{MLLM}(\Phi_{obj}(\mathbf{V}_s))), \quad \mathcal{R}_{cam} = \text{MLLM}(\Phi_{cam}(\mathbf{V}_s, \mathbf{V})). \tag{1}$$

Here, $\mathcal{R}_{obj}$ and $\mathcal{R}_{cam}$ are object and camera motion understanding, respectively. $\Phi_{obj}$ and $\Phi_{cam}$ are the corresponding visual prompting functions.

### 3.1 OBJECT REFERRING

Initially, the MLLM processes sampled frames $\mathbf{V}_s$ and the query $\mathcal{Q}$ to infer a set of semantically relevant object categories $\mathcal{C} = \{c_1, c_2, ..., c_n\}$. This inferred set $\mathcal{C}$ guides our subsequent visual perception modules for object localization and trajectory estimation. Formally, the process of obtaining tracked object trajectories $\mathcal{O}$ is defined by the composition:

$$\mathcal{O} = \mathcal{M}_{track}(\mathcal{M}_{detect}(\mathcal{I}_{s_t}, \mathcal{C}; \theta_{det}), \{\mathcal{I}_{s_j}\}_{j=t+1}^{T}; \theta_{track}), \tag{2}$$

where $\mathcal{M}_{detect}$ (Liu et al., 2024b) identifies bounding boxes for categories $\mathcal{C}$ in key frame $\mathcal{I}_{s_t}$. $\mathcal{M}_{track}$ (Ravi et al., 2024) then propagates these detections across subsequent frames $\{\mathcal{I}_{s_j}\}_{j=t+1}^{T}$ yielding trajectories $\mathcal{O}$. While direct action inference can hallucinate, robust object identification, even with initial errors, is refinable by lower-confidence detections (Chen et al., 2024c).

### 3.2 ACTION FOCUSING

Given tracked objects $\mathcal{O} = \{(\mathbf{b}_{s_t,i})_{t=1}^{T_i}\}_{i=1}^{m}$ ($\mathbf{b}_{s_t,i}$: $i$-th object's bounding box at frame $s_t$; $T_i$: trajectory length), we use a dynamic temporal aggregator $\mathcal{A}$ to derive refined spatial regions $\mathcal{B} = \{b_t\}_{t=1}^{T}$, which merge and stabilize bounding boxes against jittering. $\mathcal{A}$ adaptively adjusts its temporal aggregation window based on intra-trajectory positional variance $\mathcal{V}$. Let $\mathcal{X} = \mathcal{U}_{i=1}^{m}(\mathbf{b}_{s_{1:T_i},i})$ denote the union of bounding boxes in each frame:

$$\mathcal{B} = \mathcal{A}(\mathcal{X}, \mathcal{V}(\mathcal{X})) = \{b_t\}_{t=1}^{T}. \tag{3}$$

Here, $\mathcal{V}(\cdot)$ quantifies bounding box positional variance along a trajectory. Specifically, with low positional variance, $\mathcal{A}$ favors a union of bounding boxes over longer temporal spans; with high variance, it focuses on localized regions in shorter temporal windows. To quantify positional variance, we measure Manhattan distance between pairs of bounding box centers as $\|\text{center}(\mathbf{b}_{s_{t_1},i}) - \text{center}(\mathbf{b}_{s_{t_2},i})\|_1$.

The object motion enhancement function $\Phi_{obj}$ then applies visual prompting techniques to the original frames using these dynamically aggregated object regions:

$$\Phi_{obj}(\mathbf{V_s}) = \mathcal{F}_{VP}(\mathbf{V_s}, \mathcal{B}), \tag{4}$$

where $\mathcal{F}_{VP}$ represents our visual spotlight approach that darkens the background outside $\{b_t\}_{t=1}^{T}$ while preserving the detected objects in their original positions, enhancing focus on the relevant moving elements. In this way, our visual prompt considers object focus and smooth transition.

Table 1: **The comparison of existing motion-specific datasets with ours.** Our dataset significantly surpasses existing methodologies in both scale and annotation granularity. Furthermore, the quality of our dataset generally exceeds that of currently prevalent motion-specific datasets used for comparison.

| Dataset | #Videos | #Text | Annotation Types | Subject | Usage | Scenario |
|---|---|---|---|---|---|---|
| UCF101 (Soomro et al., 2012b) | 13K | N/A | Class Labels | Human | Action recognition | Human-centric |
| ActivityNet (Caba Heilbron et al., 2015) | 20K | N/A | Class Labels | Human | Action recognition | Human-centric |
| Kinetics-700 (Carreira et al., 2019) | 650K | N/A | Class Labels | Human | Action recognition | Human-centric |
| Charades (Sigurdsson et al., 2016) | 9K | 27K | Captions | Human, Indoor objects | Video understanding | Indoor |
| Charades-Ego (Sigurdsson et al., 2018) | 8K | 68K | Captions | Human, Indoor objects | Video understanding | Indoor |
| MotionBench-train (Hong et al., 2025) | 5K | 5K | Captions | Diverse objects, Camera | Motion understanding | Open domain |
| FavorBench-train (Tu et al., 2025) | 17K | 17K | Captions | Diverse objects, Camera | Motion understanding | Indoor |
| MotionVid − QA (ours) | 40K | 87K | Mixed QAs | Diverse objects, Decoupled camera | Fine-grained motion understanding | Open domain |
| - SFT | 35K | 80K | QAs | - | Enhance motion understanding | - |
| - DPO | 5K | 7K | Chosen/Reject QAs | - | Align with human preferences | - |

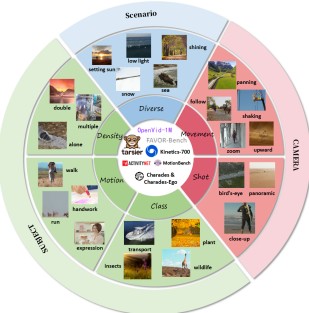
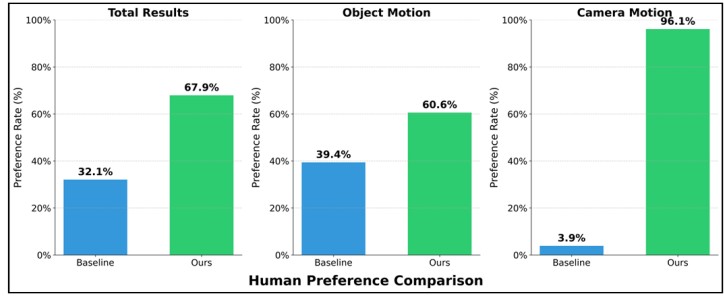

Figure 5: `MotionVid − QA`: **High-quality filtering.** Construction of a high-quality video dataset via filtration. **Extensive data distribution.** Diverse sources yield varied scenes, subjects, and camera perspectives. **Human preference comparison.** Our preference data annotation significantly surpasses baseline, particularly for camera motion (*Zoom in for best view*).

## 3.3 MOTION BLUR

To overcome the inherent limitations of existing MLLMs in perceiving inter-frame changes, particularly subtle camera motions, we introduce a dedicated Motion Blur Transformation $\mathcal{T}_{MB}$ as our camera motion core enhancement function $\Phi_{cam}$. This function, $\Phi_{cam}(\mathbf{V}, \mathbf{V}_s) = \mathbf{V}'$, operates on sampled timestamps $\{s_t\}_{t=1}^T$ using the entire video sequence $\mathbf{V} = \{\mathcal{I}_t\}_{t=1}^T$ to generate a sequence of motion-enhanced frames $\mathbf{V}' = \{\mathcal{I}'_{s_t}\}_{t=1}^T$, thereby amplifying temporal motion cues. For a given frame $\mathcal{I}_{s_t}$, the enhanced frame $\mathcal{I}'_{s_t}$ is derived through a temporally weighted aggregation of its $N$ preceding frames from the original video. This process can be formally expressed as:

$$\Phi_{cam}(\mathbf{V}, \mathbf{V}_s) = \{\mathcal{T}_{MB}(\mathbf{V}_s, N, t)\}_{t=1}^T, \quad \text{where } \mathcal{T}_{MB}(\cdot) = \sum_{k=0}^{N-1} w_k(\gamma) \cdot \mathcal{I}_{s_t-k}. \quad (5)$$

Here, $N$ is the temporal window size (zero-padding applied for $s_t - k < 1$). The temporal kernel we use has a temporally increasing trend, with $\sum_k w_k(\gamma) = 1$. This temporal aggregation within $\Phi_{cam}$ accentuates motion trajectories by inducing motion blur effects across $\mathbf{V}'$, enhancing the MLLM's capacity to perceive and interpret subtle camera movements.

## 4 MOTIONVID

In this section, our work yields two key data resources: instruction/preference subsets from public data tailored for two-stage model refinement (SFT/DPO) towards high-quality motion understanding.

**Dataset collection and processing.** `MotionVid − QA` is curated from a variety of sources to ensure multiple types of motion understanding tasks are covered, and rich scenarios ensure the diversity of the dataset, including ActivityNet (Caba Heilbron et al., 2015), Kinetics-700 (Carreira et al., 2019), Charades (Sigurdsson et al., 2016), Charades-Ego (Sigurdsson et al., 2018), Tarsier2-Recap-585K (Yuan et al., 2025), OpenVid-1M (Nan et al., 2024), and MotionBench-train (Hong et al., 2025). To ensure the quality of the videos, data processing steps we outline in Figure 2 are applied.

**SFT and DPO.** SFT aims to produce a specialized model $\pi^{\text{SFT}}$ capable of effectively capturing spatiotemporal dynamics and semantic motion patterns inherent in video data, thereby enhancing its performance on specific video understanding applications. DPO aims to simplify Reinforcement Learning from Human Feedback (RLHF) by utilizing the log-likelihood of the learning policy. Instead

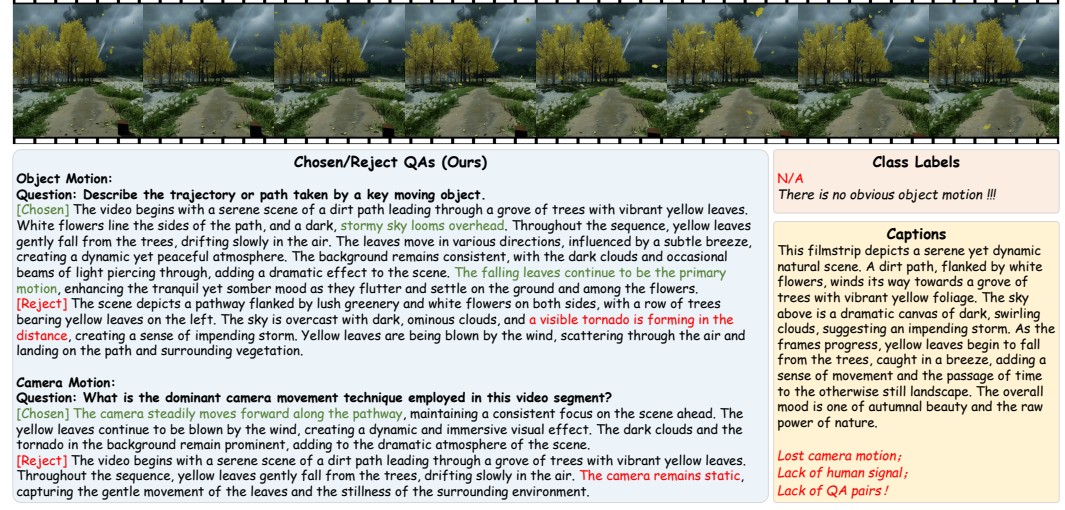

Figure 6: **A comparative visualization of** `MotionVid − QA` **against existing data.** In contrast to antecedent methodologies reliant upon class labels or captions, our approach facilitates the provision of substantially richer and more diverse informational content. Notably, even in scenarios characterized by the absence of salient principal objects, our methodology consistently yields high-quality annotations pertaining to object and camera dynamics.

of learning an explicit reward model, it implicitly expresses the reward function through pair-wise preference data $D = \{(x_i, y_i^{chosen}, y_i^{reject})\}_{i=1}^{M}$ to optimize the policy model. Let $\pi_\theta$ is the language model policy which always initialized to $\pi^{\text{SFT}}$, and $\pi_{\text{ref}}$ is also initialized from $\pi^{\text{SFT}}$. The objective function is defined as:

$$\mathcal{L}_{\text{DPO}}(\theta) = -\mathbb{E}_{(x, y^{chosen}, y^{reject}) \sim \mathcal{D}} \left[ \log \sigma \left( \beta \log \frac{\pi_\theta(y^{chosen}|x)}{\pi_{\text{ref}}(y^{chosen}|x)} - \beta \log \frac{\pi_\theta(y^{reject}|x)}{\pi_{\text{ref}}(y^{reject}|x)} \right) \right] \quad (6)$$

The DPO loss aims to maximize the reward difference between preferred and non-preferred samples, using human preference comparisons as signals (see Figure 5).

`MotionVid − QA`**: a large-scale dataset for fine-grained motion understanding.** For the pre-filtered dataset, we selected $\Theta(40K)$ clips annotated them with `MotionSight`. Using VQAScore (Lin et al., 2024) and human thresholds for categorization, high-quality clips became preference dataset candidates, low-quality ones were eliminated, and the rest formed our instruction dataset. In the SFT phase, to **enhance motion understanding capabilities via SFT**, we used `MotionSight` annotations as text data. For the preference dataset, aiming to **align fine-grained motion understanding with human preferences via DPO**. With a balance between efficiency and quality, we re-annotated this portion of the data using Tarsier2 (Yuan et al., 2025) as the baseline. High-quality preferences data was then developed by incorporating human preference signals from multiple, guided, and well-educated individuals.

Our curated dataset (Table 1, Figure 6) significantly advances fine-grained video motion understanding. Its key contributions are: (1) **Pioneering scale and scope.** The first large-scale, open-source dataset for this task, offering diverse scenes and high-quality footage. (2) **Diversity and quality.** Rigorous filtering enhances clarity (mitigating annotation hallucinations), text-video consistency and pronounced dynamics (cf. Figure 5). (3) **Hierarchical data composition.** Comprising SFT and preference subsets, it enables multi-faceted learning: general motion understanding (SFT) and refined, human-aligned fine-grained comprehension. Overall, this large-scale, high-quality, hierarchically structured resource will significantly support future model training and evaluation.

## 5 EXPERIMENTS

### 5.1 DATASETS AND COMPARISONS

**Datasets.** We evaluated our method `MotionSight` on the benchmark of MotionBench (Hong et al., 2025) and FAVOR-Bench (Tu et al., 2025) using their publicly available dev set and close-ended evaluation (encompassing six types of question-answer pairs), respectively. They are both fine-grained

Table 2: **Quantitative results on MotionBench.** We compared our `MotionSight` with both proprietary MLLMs and open-source MLLMs on MotionBench, all of which have been trained on large-scale video data. The best results of open-source methods are marked in **bold**.

| Model | # Frames | Overall | AVG. | MR | LM | CM | MO | AO | RC |
|---|---|---|---|---|---|---|---|---|---|
| *Proprietary MLLMs* | | | | | | | | | |
| Gemini 2.0 Flash (DeepMind, 2025) | 1fps | 56.1 | 52.6 | 60.9 | 57.1 | 50.9 | 74.1 | 37.6 | 35.0 |
| Gemini 1.5 Pro (Team et al., 2024) | 1fps | 51 | 48 | 51 | 52 | 54 | 67 | 40 | 22 |
| GLM-4V-Plus-0111 (ZhipuAI, 2025) | 2fps | 62.8 | 60.3 | 64.1 | 67.0 | 67.4 | 73.5 | 46.7 | 42.8 |
| *Open-source MLLMs* | | | | | | | | | |
| Oryx-34B (Liu et al., 2024d) [ICLR'25] | 64 | 49 | 47 | 48 | 52 | 44 | 65 | 42 | 32 |
| LLaVA-NeXT-Video-34B (Zhang et al., 2024a) | 32 | 48 | 44 | 53 | 45 | 36 | 66 | 39 | 23 |
| Qwen2.5VL-7B (Bai et al., 2025) | 1fps | 53.0 | 48.8 | 58.3 | 55.3 | 34.0 | 71.5 | 39.5 | 34.0 |
| Qwen2.5VL-72B (Bai et al., 2025) | 1fps | 58.3 | 54.3 | 64.0 | 60.3 | 48.6 | 73.2 | 46.8 | 33.0 |
| InternVL3-8B (Zhu et al., 2025a) | 16 | 58.1 | 53.7 | 65.1 | 63.0 | 47.8 | 74.1 | 39.7 | 32.3 |
| InternVL3-78B (Zhu et al., 2025a) | 16 | 61.5 | 57.6 | 67.2 | 63.9 | 55.8 | 78.1 | 44.9 | 35.8 |
| TE Fusion (Hong et al., 2025) [CVPR'25] | 16 | 58 | 54 | 64 | 59 | 51 | 69 | 41 | **39** |
| Qwen2.5VL-7B + MotionSight | 1fps | 55.6 | 52.2 | 59.7 | 58.1 | 48.3 | 73.6 | 40.1 | 33.5 |
| InternVL3-78B + MotionSight | 16 | **63.0** | **59.3** | **68.5** | **65.4** | **58.7** | **78.6** | **47.6** | 37.0 |

motion-level benchmarks encompassing a wide range of video types. We report accuracy for each problem type, with **Overall** representing accuracy across all problems. We also calculate **AVG.** as the average accuracy across question categories, giving equal weight to each problem type regardless of sample distribution. It's worth noting that our `MotionSight` is an untrained zero-shot enhancement scheme, while `MotionChat` is trained on the `MotionVid − QA` dataset.

**Quantitative evaluation.** Table 2 and Table 3 presents the quantitative results on MotionBench and FAVOR-Bench, respectively. Our `MotionSight` consistently enhances the performance of base MLLMs. When using Qwen2.5VL as the backbone, our method achieves a 3.4% improvement on MotionBench and a 3.0% improvement on FAVOR-Bench in category average (AVG.), while **camera motion** improves by 14.3% on MotionBench. Effective performance improvements were also observed in metrics tightly coupled with **object motion**, including MR,

Table 3: **Quantitative results on FAVOR-Bench.** We selected representative MLLMs as baselines for comparison. We computed the OM (object motion) metric by averaging all metrics excluding the CM (camera motion) metric in FAVOR-Bench.

| Model | # Frames | Overall | AVG. | OM | CM |
|---|---|---|---|---|---|
| *Proprietary MLLMs* | | | | | |
| GPT-4o (Hurst et al., 2024) | 1fps | 42.1 | 43.1 | 44.5 | 36.0 |
| Gemini-1.5-Pro (Team et al., 2024) | 1fps | 49.9 | 50.7 | 52.6 | 41.6 |
| Claude-3.7-Sonnet (Anthropic, 2025) | 1fps | 43.7 | 44.0 | 45.0 | 39.1 |
| *Open-source MLLMs* | | | | | |
| LLaVA-NeXT-Video-34B (Liu et al., 2024a) | 8 | 30.4 | 32.6 | 33.2 | 29.6 |
| VideoLLaMA3-7B (Zhang et al., 2025) | 1fps | 41.5 | 41.5 | 43.4 | 31.5 |
| Qwen2.5VL-7B (Bai et al., 2025) | 1fps | 42.3 | 41.6 | 43.7 | 30.9 |
| InternVL3-78B (Zhu et al., 2025a) | 16 | 52.8 | 52.6 | 56.2 | 34.3 |
| Qwen2.5VL-7B + MotionSight | 1fps | 45.1 | 44.1 | 45.3 | **38.1** |
| InternVL3-78B + MotionSight | 16 | **53.8** | **53.5** | **56.8** | 37.1 |

LM, MO and AO. Furthermore, InternVL3-78B augmented with `MotionSight` demonstrates a significant performance enhancement, achieving state-of-the-art results among open-source models like TE Fusion (Hong et al., 2025) and exhibits strong competitiveness against leading proprietary models like GLM-4V-Plus-0111 (ZhipuAI, 2025).

**Qualitative Evaluation.** As shown in Figures 7 and 9, the visual spotlight of `MotionSight` enhances the model's ability to focus more on motion information, while the motion blur method significantly enables the model to perceive changes in camera motion in videos. Our approach to decoupling object motion and camera motion leads to a significant improvement in accuracy. Figure 6 presents our dataset, which we believe will make substantial contributions to the research community.

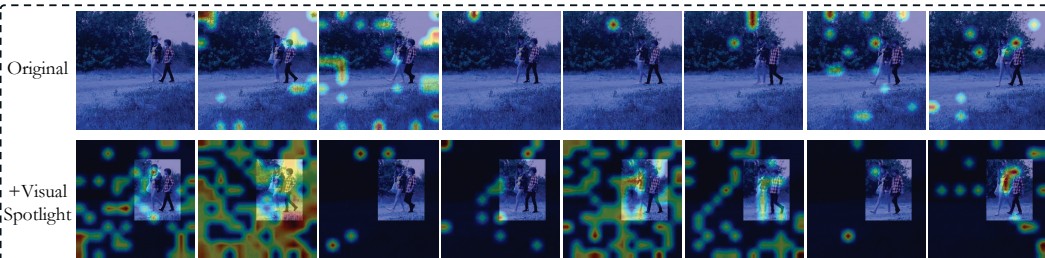

Figure 7: **The difference between using visual spotlight and the original MLLM.** We used Grad-CAM and selected the same layer for gradient computation. After incorporating the visual spotlight, the model pays more attention to the core region. Prompt: *"What are the people doing?"*.

**MotionChat.** We conducted a fine-tuning experiment on our self-built `MotionVid − QA` dataset to demonstrate its core effectiveness. Based on Qwen2.5VL-7B, we first performed a full-parameter SFT training, followed by DPO training on the SFT model, with the vision module frozen during DPO training. As shown in Table 4, after the two-stage fine-tuning on `MotionVid − QA`, our `MotionChat` significantly outperformed the original model. Furthermore, it achieved an overall ac-

Table 4: **Quantitative results for `MotionChat` based on Qwen2.5VL-7B across different training stages.** We analyzed the impact of different training strategies by selectively including or excluding our SFT and preference datasets on FAVOR-Bench. Green areas indicate best performance. "✔" indicates the presence of a training stage, while "✘" indicates its absence. "+ShareGPT4Video" denotes that we fine-tuned Qwen2.5VL-7B solely on the ShareGPT4Video dataset, demonstrating the superior quality of our data.

| SFT | DPO | Overall | AVG. | AS | HAC | SAD | MAD | CM | NSM |
|---|---|---|---|---|---|---|---|---|---|
| ✔ | ✔ | 48.3 | 46.9 | 49.6 | 54.9 | 45.9 | 55.1 | 32.1 | 43.8 |
| ✔ | ✘ | 45.8 | 44.5 | 47.3 | 51.6 | 43.7 | 52.3 | 30.1 | 42.2 |
| ✘ | ✘ | 42.3 | 41.6 | 41.6 | 46.7 | 43.5 | 46.3 | 30.9 | 40.6 |
| +ShareGPT4Video | | 43.8 | 42.3 | 45.6 | 45.8 | 44.9 | 49.2 | 28.9 | 39.1 |

curacy of 48.3% on FAVOR-Bench, which is comparable to the motion understanding ability of Qwen2.5VL-72B.

**General Video Understanding.** To verify the broader impact of our approach, we tested VideoMME. The results, shown in Table 5 and Figure 8, indicate that `MotionSight` improves performance on several general-purpose tasks, particularly those related to object motion, and does not exhibit significant global information loss. Our method outperforms the baseline on most metrics and achieves performance comparable to the LLaVA-OneVision-72B model. We attribute this enhancement to the visual spotlight, which helps the model focus on task-relevant regions, thereby yielding consistent performance gains. To further investigate the generalizability and extended value of our approach, we conducted additional experiments with `MotionSight` on MVBench (Li et al., 2024a), TOMATO (Shangguan et al., 2024), STI-Bench (Li et al., 2025) and TempCompass (Liu et al., 2024c), as shown in Table 7. Notably, results also show that our decoupling strategy brings additional gains in spatial perception and understanding.

## 5.2 ABLATION STUDY

**Object motion understanding.** We evaluate visual prompting strategies to improve object motion understanding. As detailed in Table 6, our proposed visual spotlight yields the highest average object motion score (OM AVG.). Background blur, however, negatively impacted performance, contrasting its effectiveness in static image prompting (Yang et al., 2023b). We attribute this failure to blurred object boundaries, increasing the demand for robustness and misleading MLLMs. Other visual prompts, including object crop, object motion blur (applied solely to the object mask) and pose estimation (applied to the entire video), also provided marginal or negative impacts. These findings underscore the efficacy of visual spotlight in directing model attention to pertinent object movements.

Table 5: **Evaluation on VideoMME.** We present the core general-purpose tasks.

| Model | Overall | Temporal Perception | Action Recognition | Object Recognition | Temporal Reasoning | Spatial Reasoning | Action Reasoning | Information Synopsis |
|---|---|---|---|---|---|---|---|---|
| Qwen2.5VL-7B | 73.7% | 83.3% | 77.1% | 74.4% | 76.9% | 74.1% | 68.1% | 86.6% |
| +Ours | **76.0%** | **88.9%** | **80.2%** | **76.2%** | **84.6%** | **77.8%** | **70.2%** | **87.8%** |

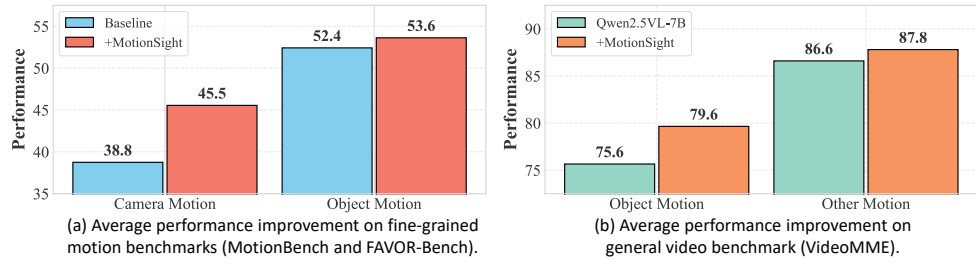

(a) Average performance improvement on fine-grained motion benchmarks (MotionBench and FAVOR-Bench).

(b) Average performance improvement on general video benchmark (VideoMME).

Figure 8: We compiled the metrics related to object motion, camera motion, and other motion from the benchmark and averaged the experimental model and task metrics. Other motion refers to tasks with low correlation to object motion, for which we also used the visual spotlight to focus on core regions. Our method shows significant advantages after decoupling motion.

Table 6: **Experiments of several visual prompt methods specialized for motion decoupling on MotionBench**. Green areas indicate best performance and red areas show lowest scores.

| Method | Object Motion | | | | | | Camera Motion |
|---|---|---|---|---|---|---|---|
| | OM AVG. | MR | LM | MO | AO | RC | CM AVG. |
| Qwen2.5VL-7B (Bai et al., 2025) | 51.7 | 58.3 | 55.3 | 71.5 | 39.5 | 34.0 | 34.0 |
| **+ Visual Spotlight** | 53.0 | 59.7 | 58.1 | 73.6 | 40.1 | 33.5 | - |
| + Object Crop | 52.5 | 59.0 | 55.9 | 71.3 | 40.1 | 36.5 | - |
| + Background Blur (Yang et al., 2023b) | 49.3 | 56.2 | 53.9 | 65.4 | 39.9 | 31.3 | - |
| + Object Motion Blur | 50.2 | 60.2 | 52.5 | 68.8 | 40.4 | 29.3 | - |
| + Pose Estimation | 50.6 | 56.4 | 54.6 | 70.1 | 40.1 | 32.0 | - |
| **+ Global Motion Blur** | 50.8 | 57.2 | 57.0 | 69.1 | 39.3 | 31.5 | 48.3 |
| + API Prompting (Yu et al., 2024) | 51.4 | 57.9 | 55.5 | 73.6 | 39.3 | 30.5 | – |

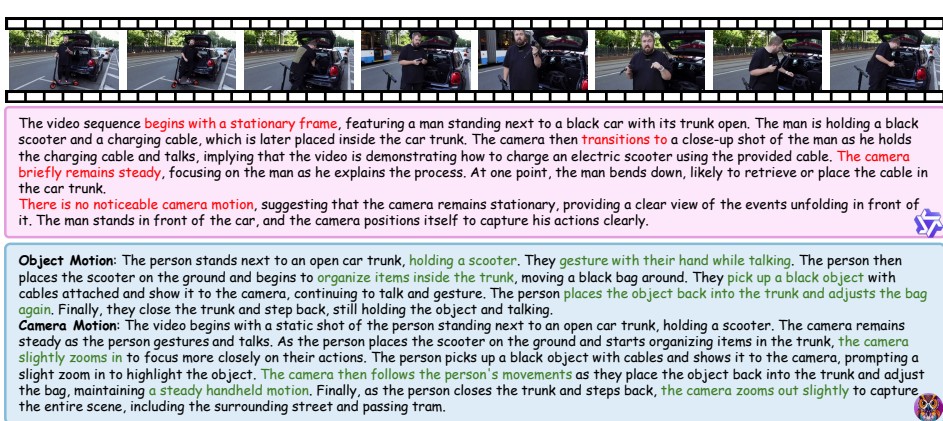

Figure 9: **Qualitative example of** `MotionSight`. The upper panel represents the baseline, while the lower panel shows the result enhanced by our method, which possesses fine-grained inter-frame difference perception, enabling precise capture of subtle motions.

**Camera motion understanding.** For camera motion, our primary evaluation focused on applying motion blur to the entire video frame, also referred to as global motion blur. Table 6 shows that our motion blur synthesis approach facilitates the model's perception of subtle inter-frame differences, thereby yielding a substantial improvement, significantly outperforming the baseline. The results also demonstrate the effectiveness of our decoupled object and camera motion method. Please refer to Section B in the appendix for more ablation studies.

Table 7: Performance of `MotionSight` on more video understanding benchmarks.

| Method | MVBench | TOMATO | STI-Bench | TempCompass |
|---|---|---|---|---|
| Qwen2.5-VL-7B | 60.5 | 26.8 | 31.6 | 68.5 |
| + MotionSight | **64.4** | **27.3** | **36.5** | **71.3** |

## 5.3 MORE APPLICATIONS

**Applications to the video generation.** The `MotionVid − QA` can not only be used for MLLM fine-tuning but also support video generation, enhancing motion quality in generated videos. We constructed video-text pairs from the large-scale, high-quality data in the SFT subset, and conducted fine-tuning on Wan2.1-T2V-1.3B, followed by evaluation on VBench. As shown in Table 8 and Figure 10, after our efficient fine-tuning, the model achieves significant improvements in motion-related generation, which can be attributed to the high quality and rich fine-grained motion content of our dataset.

**Self-improving VLM pipeline.** We tested the further application of `MotionSight` on top of `MotionChat` and found that it can further improve the model's performance, as shown in Table 9. Therefore, `MotionSight` has the potential to serve as a self-improving optimization approach. This introduces a novel paradigm for unique self-enhancing motion understanding capabilities: we can

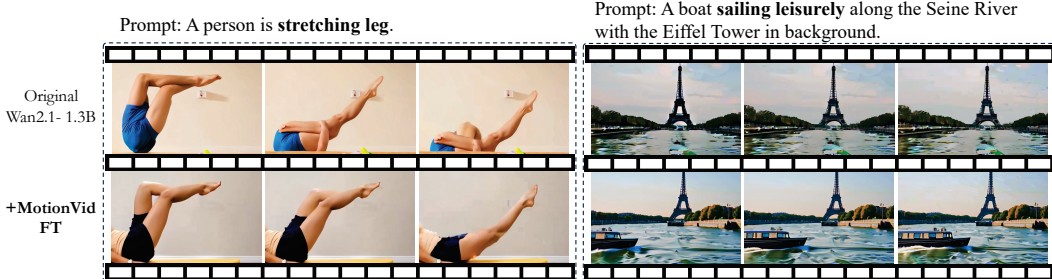

Figure 10: Qualitative example of our fine-tuned Wan2.1-T2V-1.3B.

Table 8: Fine-tuning results on `MotionVid − QA` for video generation.

|  | Human Action | Motion Smoothness | Dynamic Degree | Temporal Style |
|---|---|---|---|---|
| Wan2.1-T2V-1.3B | 64.2 | 97.9 | 47.7 | 20.4 |
| + MotionVid − QA | **71.0** | **98.3** | **54.3** | **21.5** |

Table 9: Results after applying `MotionSight` on top of the fine-tuned `MotionChat` model.

| Method | Overall | AVG. | AS | HAC | SAD | MAD | CM | NSM |
|---|---|---|---|---|---|---|---|---|
| MotionChat | 48.3 | 46.9 | 49.6 | 54.9 | 45.9 | 55.1 | 32.1 | 43.8 |
| MotionChat + MotionSight | 49.7 | 48.3 | 50.7 | 55.3 | 46.0 | 56.0 | 38.2 | 43.8 |

leverage "MotionChat+ MotionSight" as a more powerful *teacher* to generate next-generation datasets, thereby continuously pushing the ceiling of VLMs in fine-grained motion understanding.

## 6   DISCUSSIONS ON NOVELTY AND VALUE

We are the first in the motion understanding field to propose a methodology that decouples object motion from camera motion. The challenge of fine-grained motion understanding lies in the difficulty MLLMs have in perceiving subtle inter-frame changes:

- Existing video understanding models perceive motion *holistically* and struggle to separate object motion from camera motion. In contrast, our specially designed *decoupling* strategy makes it easier for the model to perceive subtle camera motion and object motion separately.

- Simply applying *static* image prompting methods commonly used in previous work to videos can lead to misunderstandings (Figure 3). Our designed algorithm possesses *dynamic* stability, which helps enhance the MLLM's ability to perceive fine-grained motion.

- Existing datasets are *limited* to indoor or human-centric scenarios with *simple labels* (Table 1). In contrast, we constructed the *first* large-scale open-source fine-grained video motion understanding dataset, `MotionVid − QA`, which includes *open-domain* motion and *fine-grained labels* such as decoupled motion and preference question pairs.

Moreover, `MotionChat` offers strong practical efficiency (as shown in Table 21): it outperforms the 72B baseline while being approximately 5-6× faster and far more memory-efficient.

## 7   CONCLUSIONS

In this work, we address the challenge of fine-grained video motion understanding, a task that has been relatively underexplored despite the advancements in MLLMs. We introduce `MotionSight`, a novel zero-shot approach that decouples object motion from camera motion. By employing visual spotlight and motion blur, it enhances the ability of MLLMs to perceive subtle motion cues. Furthermore, we curate `MotionVid − QA`, the first large-scale open-source dataset designed for fine-grained video motion understanding. Our experiments demonstrate the effectiveness of `MotionSight` and `MotionVid − QA` to facilitate future research in this domain.

ACKNOWLEDGEMENT

This work was supported by the Gusu Innovation and Entrepreneur Leading Talents: No. ZXL2024362, Natural Science Foundation of Jiangsu Province: BK20241198, and Natural Science Foundation of China: No. 62406135.

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

## A   IMPLEMENTATION DETAILS

### A.1   EXPERIMENTAL SETTINGS

We performed all experiments of `MotionSight` on 8 NVIDIA 4090 GPUs with 48GB of memory each. For GroundingDINO, the box threshold and text threshold for post-processing grounded object detection are both set to 0.25. For SAM 2, the mask dictionary model uses an IOU threshold of 0.8 for updating masks. In motion blur part we use decay factor $\gamma = 0.65$ and temporal window $N = 7$. For action focusing we use darken factor $\beta = 0.9$. To address potential detection loss due to frequent object entry and exit (e.g., FAVOR-Bench Tu et al. (2025)), we perform re-detection using $\mathcal{M}_{detect}$ at fixed intervals $\Delta t$, ensuring comprehensive object capture. The weighted mean (temporal kernel) in Sec 3.3 is implemented as follows:

$$w_{N-1-k}(\gamma) = \gamma^k \cdot \prod_{j=k+1}^{N-1} (1 - \gamma^j) \tag{7}$$

For the fine-tuning experiments, we utilized 32 A100 GPUs with 80GB of memory each. The SFT process was conducted on the Qwen2.5-VL-7B-Instruct model, employing a global batch size of 128. During SFT, the vision tower, LLM, and merger components were all trainable. During DPO training, we exclusively trained the LLM part.

## A.2 PROMPT TEMPLATE

We present our prompt details. For the Object Referring stage, we use the prompt shown in the upper image to enable the MLLM to locate the most critical objects based on the video and question content. We utilize the concept of action groups , allowing the MLLM to identify relevant objects—and even their components—in a fine-grained manner.

The code below defines a key part of our final configuration template. Based on enhanced inputs such as Action focus or motion blur, we route the input into the MLLM using a routing mechanism. In this template, `description_for_video_type` refers to descriptions tailored for different video types.

```
video_descriptions = {
    'original': 'Original video:\n',
    'spotlight': 'Spotlight video:\n',
    'motion_blur': 'Original video with motion blur to more
    ↪    clearly determine the type of motion (such as whether the
    ↪    camera is moving, as one frame combines information from
    ↪    multiple frames. If static objects in the background
    ↪    appear noticeably blurry, there is a good chance that the
    ↪    camera is moving!):\n'
}
```

---

**Action group analyze.**

```
<video><time_info>
```

I have a question: "`<question>`". I need you to analyze the above question step by step. In this step, you don't need to directly answer the question.

Please provide your response in the following JSON format without any comment:

```
{
    "action_objects":  ["object1", "object2", ...],
}
```

For the "action_objects" field, provide a list of strings, each describing a specific entity that is involved in the main action or motion. Each entity should be a single object or a group of objects. For example, if the question is about a person eating, include both the person and the rice bowl. If the question is about object motion, make sure to include both the moving objects (actors/performers) and the objects they interact with or affect. You can also provide fine-grained components of larger objects when relevant. Each string represents a different object. All items must be physical entities that can be visually identified, not abstract concepts. Only keep the moving objects that are highly relevant to the question and reduce the background objects. You must provide at least one action_object.

---

**Config Template.**

```
<video><time_info>
```

The video contains {len_frames} frames sampled at {sec} seconds.

{description_for_video_type_1}
Frame{frame_num}: {IMAGE_TOKEN}

...
Frame{frame_num}: {IMAGE_TOKEN}
{description_for_video_type_2}

---

Frame{frame_num}: {IMAGE_TOKEN}

...

Frame{frame_num}: {IMAGE_TOKEN}

Here is the question: "`<question>`".

Reply based on the above information. Answer only the answer letter without showing your process.

# B  MORE EXPERIMENTS

## B.1  APPLICATIONS TO GENERAL VIDEO UNDERSTANDING

We conducted additional evaluations with Qwen2.5VL-7B being the backbone on the TempCompass benchmark to prove the general effectiveness of `MotionSight` and impact of global information.

Table 10: Evaluation results on the main TempCompass benchmark.

| Model | Avg | Action | Attribute Change | Direction | Order | Speed |
|---|---|---|---|---|---|---|
| Qwen2.5VL-7B | 68.50 | 94.40 | 76.20 | 50.42 | 71.62 | 50.65 |
| +Ours | **71.32** | **94.72** | **79.03** | **52.55** | **74.47** | **58.40** |

Table 11: Performance on the captioning subset of TempCompass.

| Model | Avg | Action | Attribute Change | Direction | Order | Speed |
|---|---|---|---|---|---|---|
| Qwen2.5VL-7B | 54.6 | 92.7 | 68.5 | 32.8 | 51.2 | 29.9 |
| +Ours | **64.0** | **93.2** | **77.1** | **38.1** | **67.0** | **47.4** |

Table 12: Breakdown on TempCompass's fine-grained captioning sub-dimensions. AS: Absolute Speed; CM: Camera Motion; CGA: Coarse-grained Action; CLC: Color & Light Change; CC: Combined Change; FGA: Fine-grained Action; OC: Other Change; RS: Relative Speed; SSC: Size & Shape Change.

| Model | AS | CM | CGA | CLC | CC | FGA | Other | OC | RS | SSC |
|---|---|---|---|---|---|---|---|---|---|---|
| Qwen2.5VL-7B | 43.9 | 15.0 | 95.1 | 76.9 | 70.0 | 90.1 | 51.2 | 66.7 | 22.7 | 63.1 |
| + MotionSight | **62.9** | **31.7** | **95.6** | **87.5** | **75.0** | **90.6** | **67.0** | **75.0** | **39.5** | **72.2** |

For TempCompass (as shown in Table 12, Table 10, Table 11), we compared our method with Qwen2.5VL-7B under the same settings, using "gpt-4o-mini-2024-07-18" as the LLM evaluator for TempCompass. The results show that our method significantly outperforms the baseline on TempCompass, demonstrating strong advantages even in more general temporal understanding tasks. Notably, we observed that our method achieves far superior performance on subtasks with higher output freedom, such as captioning. This reflects a deeper enhancement in motion understanding: objective questions are essentially simple discriminative tasks, often requiring only the judgment of isolated motion facts. In contrast, captioning demands more profound and fine-grained descriptions, which aligns well with the core design of `MotionSight` for refined motion understanding. In summary, **our approach has greater universality and potential.**

## B.2  MORE VALIDATION OF DATASET QUALITY AND GENERALIZATION CAPABILITY

To further verify the generalization capability of MotionVid-QA on stronger models, we also fine-tuned the Qwen3-VL-8B-Instruct model. Notably, the original Qwen3-VL-8B model (overall score of 53.0) has already surpassed the InternVL3-78B model (52.8 in Table 3). As shown in Figure 11 and Table 13, **the results demonstrate that even on already strong models, our method can still significantly improve performance, which validates the excellent generalizability and complexity of our dataset.**

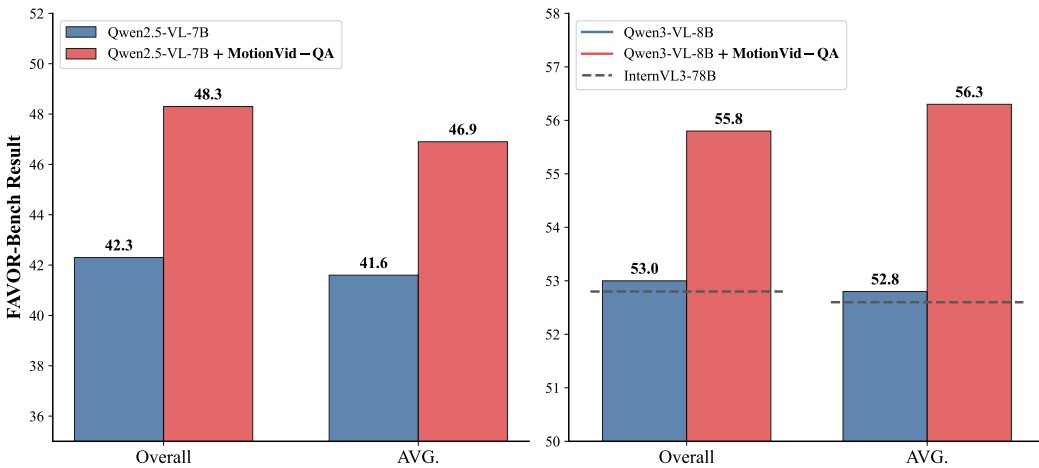

Figure 11: The training results on Qwen2.5-VL-7B and Qwen3-VL-8B using our dataset show that our data can also enhance even stronger models, demonstrating its high quality, high complexity, and strong generalization capability.

Table 13: Results of Fine-tuning Qwen3-VL-8B-Instruct with MotionVid-QA on FAVOR-Bench

| Method | Overall | AVG. | AS | HAC | SAD | MAD | CM | NSM |
|---|---|---|---|---|---|---|---|---|
| Qwen3-VL-8B | 53.0 | 52.8 | 52.9 | 58.5 | 52.7 | 56.4 | 41.7 | 54.7 |
| +MotionVid FT | **55.8** | **56.3** | **55.4** | **64.3** | **53.0** | **61.5** | **42.5** | **60.9** |

After fine-tuning with MotionVid-QA, the overall accuracy increased by 2.8%, and the average accuracy increased by 3.5%. Our fine-tuned 8B model (55.8%) also significantly outperforms the InternVL3-78B model (52.8%). The substantial gains on complex tasks such as HAC (+5.8%) and NSM (+6.2%) further validate that the filtered dataset retains challenging fine-grained motion semantics, which remain applicable even to powerful baseline models.

## B.3    MORE ABLATION STUDIES ON ROBUSTNESS

To clarify that our method demonstrates strong robustness to hyperparameters, we conducted ablation studies on essential parameters: the degree of background darkening in the bounding-box-based spotlight (from slight dimming to completely black background), temporal window size, and the decay factor discussed in Section 3.3.

To evaluate the robustness of background darkening in the visual spotlight, we performed denser sampling within the selected parameter range from 0.1 to 1.0. The results are shown in Table 15. We perform separate analyses for the temporal window size and the decay factor to assess their individual robustness on model performance, as shown in Table 16 and Table 17.

Overall, the ablation results show that our method significantly outperforms the baseline under different parameter settings. Thus, **our method** `MotionSight` **has strong robustness.** We supplemented more visualizations of `MotionSight`, as shown in the Figure 14.

Table 15: Ablation study on the degree of background darkening.

| Darken | Baseline | 0.1 | 0.5 | 0.8 | 0.85 | 0.9 | 0.95 | 1 |
|---|---|---|---|---|---|---|---|---|
| OM AVG. | 51.7 | 52.5 | 52.5 | 51.9 | 52.1 | **53.0** | 52.1 | 52.0 |

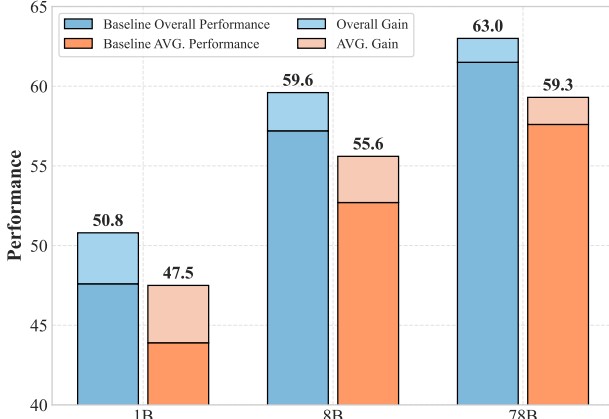

Figure 12: Ablation study on model scalability.

Table 14: Ablation study of different input methods.

| Model | OM AVG. |
|---|---|
| Qwen2.5VL-7B | 51.7 |
| Our MotionSight | **53.0** |
| Direct Input Coords | 51.8 |

Table 16: Ablation study on temporal window size ($N$ in Equation 5), with fixed decay factor: 0.65.

| Temporal Window | Baseline | 5 | 7 | 9 | 11 |
|---|---|---|---|---|---|
| CM | 34.0 | 48.1 | **48.3** | 46.0 | 41.0 |

Table 17: Ablation study on the decay factor ($\gamma$ in Equation 5), with fixed temporal window: 7.

| Decay Factor | Baseline | 0.4 | 0.65 | 0.85 |
|---|---|---|---|---|
| CM | 34.0 | 46.2 | **48.3** | 45.2 |

Table 18: Performance under different frame sampling parameters.

| Method | Baseline | 1 fps | 8 frames | 4 frames |
|---|---|---|---|---|
| MotionSight | 53.0 | 55.6 | 55.4 | 54.1 |

Table 19: Performance under different max_pixels parameters.

| Method | Baseline | 360p | 720p | native |
|---|---|---|---|---|
| MotionSight | 53.0 | 55.6 | 55.8 | 55.9 |

## B.4 ABLATION STUDIES ON MORE PARAMETERS

To further investigate the robustness of our method, we conducted experiments with a wider range of parameters. The results show that **Our method exhibits strong robustness to the parameter choices of the selected MLLM.**

**Robustness to frame sampling parameters.** Sampling at 1 fps, 8 frames, or 4 frames all yield stable improvements over the baseline, as shown in Table 18.

**Robustness to max_pixels resolution parameters.** For different max_pixels settings in the MLLM (where "native" indicates no scaling), our method consistently and stably outperforms the baseline, as shown in Table 19.

**Architectural robustness.** The two architectures used in our experiments (Qwen2.5-VL and InternVL3) represent the two mainstream typical MLLM architectures, which in itself demonstrates the robustness of our approach to different selected models.

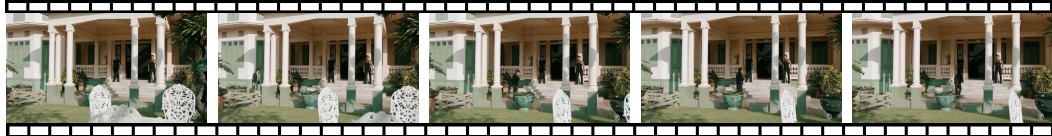

In the video, there are two individuals standing on the porch of a house. One person is walking towards the left side of the frame, while the other remains stationary near the doorway. The porch is adorned with several potted plants and decorative items, including a large blue pot and some ornate white furniture. The scene appears to be set during the daytime, as indicated by the bright lighting and shadows cast by the objects and people. There are no significant actions being performed by the objects themselves; they are static elements in the environment.

The video begins with a person in a black outfit walking towards the house entrance from the left side of the frame. As they approach, an individual standing at the doorway raises their hand in a gesture. The first person continues to walk forward, and upon reaching the steps, they pause briefly before entering the house. Throughout the sequence, the individuals remain mostly stationary except for the movement of the person walking and the subtle gestures of the individual at the doorway. The surrounding environment, including the plants and chairs, remains unchanged.

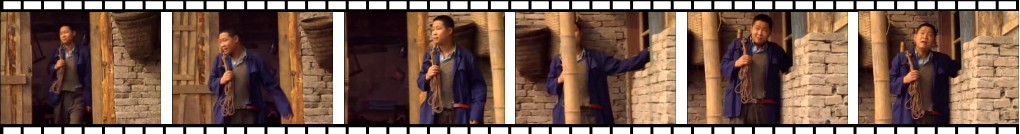

In the video, a person is seen exiting a rustic building, holding a coiled rope and a bundle of sticks or firewood. The individual appears to be dressed in traditional attire, suggesting a rural or agricultural setting. As they step out, they seem to be engaged in some form of work or activity related to farming or outdoor labor, given the tools they are carrying. The environment around them includes stone walls and wooden structures, reinforcing the impression of a countryside or village location. The person's movements suggest they might be heading towards a task that requires these items, such as tending to livestock or preparing for a fire.

The video begins with a person exiting a doorway, holding a rope in their right hand and a small object in their left hand. As they step out, they walks forward, swinging slightly as they move. Their left arm extends outward, pointing or gesturing towards something off-screen. Throughout the sequence, the person maintains a steady pace, continuing to gesture with their left arm while holding the rope in their right hand. The background remains consistent, featuring the wooden and brick structure of the house, with no significant changes in the scene or additional interactions.

Figure 13: Quantitative results between baseline and our fine-tuned models trained using both SFT and DPO. For each case, the upper one is the Qwen2.5VL-7B baseline, and the lower one is our model after fine-tuning.

Table 20: Performance across MLLM models of different parameter scales.

| InternVL3 | Overall | AVG. |
|---|---|---|
| 1B | 47.6 | 43.9 |
| +Ours | 50.8 (+3.2) | 47.5 (+3.6) |
| 8B | 57.2 | 52.7 |
| +Ours | 59.6 (+2.4) | 55.6 (+2.9) |
| 78B | 61.5 | 57.6 |
| +Ours | 63.0 (+1.5) | 59.3 (+1.7) |

### B.5 SCALABILITY OF MotionSight

We conducted experiments based on InternVL3, ranging from 1B to 78B parameters, demonstrating that our method exhibits **good scalability across popular model sizes**. The results on MotionBench are shown in Figure 20 and Table 20. The lighter portion within each bar represents the performance improvement of our method over the baseline model.

### B.6 QUALITATIVE RESULTS OF MotionChat ON THE DATASET

As shown in Figure 13, MotionChat exhibits enhanced fine-grained motion perception, more accurately interpreting complex motion narratives compared to the baseline model. **Our fine-tuned model, trained on our dataset, demonstrates superior fine-grained motion perception capabilities and outperforms the baseline.**

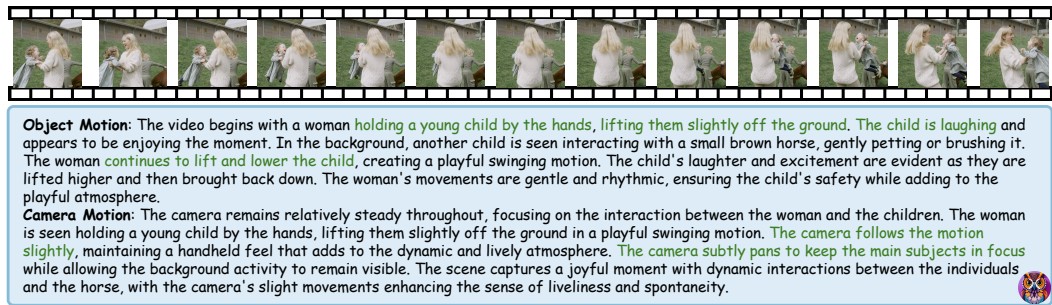

Figure 14: Another qualitative example of `MotionSight`.

## C  VISUAL SPOTLIGHT'S MOTIVATION

Visual spotlight is designed to help the model better focus on the most crucial regions of temporal changes when processing video frames. In scenarios like TV shows and movies, important moving subjects are often highlighted while the background is dimmed. Additionally, some performance shows often use techniques such as stage spotlights. MLLMs inevitably encounter such data during training (e.g., datasets like Panda-70M, which contain a large amount of TV program content). Also, using the visual spotlight method can indirectly help MLLMs concentrate their visual attention on core motion regions, intuitively and effectively improving the model's ability to capture motion information.

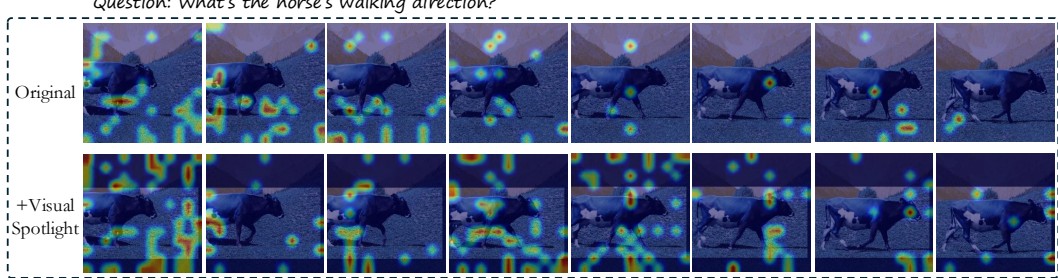

Figure 15: Another Grad-CAM visualization of our visual spotlight method after vision module. Despite using a smaller darken factor, the model can still provide more attention to moving objects.

Figure 7 and Figure 15 shows the Grad-CAM visualization results after applying our visual spotlight method to Qwen2.5VL-7B. The first figure uses a higher darken factor, while the second figure uses a slightly lower one. As can be seen, our method allows the model to focus more on the motion-related visual parts relevant to the question across different darken factors, thereby improving its fine-grained motion understanding capabilities.

Furthermore, we have implemented a comparative experiment where bounding box coordinates are directly provided as prompts, as shown in Table 14. The results show that although providing bounding box coordinates is also an intuitive prompting method, it does not lead to noticeable performance improvement. This is because it is challenging for the MLLMs to precisely apply the coordinate information to the video, and coordinates require significantly longer text context, demanding higher parsing capabilities from the model. This phenomenon further **validates the reasonableness and effectiveness of our method** in terms of engineering implementation and model adaptation.

## D  DISCUSSIONS ON COMPUTATIONAL OVERHEAD

Performance is a common engineering challenge for many current training-free enhancement modules and workflows, such as in InstanceCap (Fan et al., 2024a) [CVPR'25], VideoTree (Wang et al., 2025b) [CVPR'25], and VideoAgent (Fan et al., 2024b) [ECCV'24].

Table 21: Taking 1080p input as an example, performance testing across different frame numbers. Each setting is tested 5 times and the average is reported.

| Method | FAVOR-Bench | 8 frames, 1080p | | 16 frames, 1080p | | 32 frames, 1080p | |
|---|---|---|---|---|---|---|---|
| | | Time | Mem | Time | Mem | Time | Mem |
| Qwen2.5-VL-7B | 42.3 | 8.15s | 27.8GB | 15.5s | 28.2GB | 31.0s | 29.0GB |
| Qwen2.5-VL-72B | 48.1 | 39.2s | 320.8GB | 93.1s | 322.9GB | 207.7s | 330.4GB |
| MotionChat-7B (Ours) | **48.3** | 7.14s | 27.8GB | 13.5s | 28.2GB | 32.1s | 29.0GB |

To address this, we propose an alternative solution that achieves superior performance on fine-grained motion understanding tasks with a computational cost equivalent to the native Qwen2.5VL-7B model. This solution involves fine-tuning the model on our meticulously curated dataset. Detailed computational time and memory overhead are shown in the Table 21, demonstrating that our approach not only outperforms the 72B results in effectiveness but is also approximately 5-6× faster. Meanwhile, we have optimized the MotionSight pipeline, reducing the average increase in inference latency to less than 75%.

In summary, for users who seek the highest performance, we've optimized the MotionSight pipeline, which is a specific and valuable workflow improvement; for users who prioritize efficiency, we offer MotionChat—an end-to-end model fine-tuned on a new dataset, with a computational cost comparable to the base model.

# E    DATASET FILTER AND CURATION FOR MotionVid − QA

In this section, we focus on the methodological aspects and specific steps involved in filtering and curating the instruction and preference subsets for SFT and DPO, providing supplementary details to the dataset construction process of MotionVid − QA described in Section 4. Figure 16 showcases the examples we filtered.

## E.1    INITIAL DATA COLLECTION AND PRE-PROCESSING

MotionVid − QA was aggregated from multiple sources (Section 4), resulting in an initial set of video clips, denoted as $C_R$. These clips underwent an initial data processing pipeline, $\mathcal{P}_{initial}$. The pipeline $\mathcal{P}_{initial}$ filters the raw clips $C_R$ such that only clips satisfying specific quality metrics are retained. Specifically, a clip $c \in C_R$ is included in the pre-filtered set $C_P$ if its optical flow score $s_f(c)$ is above a threshold $\tau_f$ AND its clarity score $s_c(c)$ is above a threshold $\tau_c$. The set of pre-filtered clips is thus defined as:

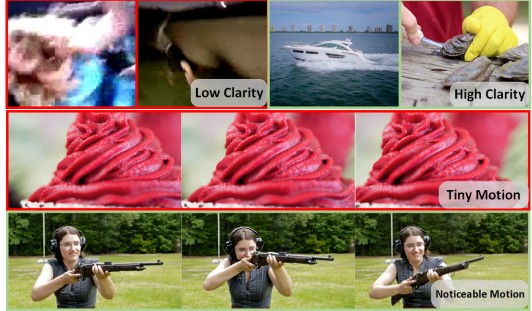

$$C_P = \{c \in C_R \mid s_f(c) > \tau_f \wedge s_c(c) > \tau_c\} \tag{8}$$

Figure 16: **More examples of** MotionVid − QA**.** We filtered out low-quality videos and kept the high-quality ones. Our MotionVid − QA includes varied scenes, subjects, and camera perspectives.

## E.2    DETAILED FILTERING AND SUBSET CREATION FOR SFT AND DPO

The methodology for creating SFT and DPO subsets from the pre-filtered set $C_P$ is introduced in Section 4 of the main paper.

**Initial selection and annotation.**    A subset of clips was chosen from $C_P$ for annotation with MotionSight. Let $C_A$ denote the set of successfully annotated clips.

**Annotation quality-based categorization.**    Each clip $c \in C_A$ was evaluated using VQAScore Lin et al. (2024). To ensure the rationality of the chosen VQAScore thresholds for categorization (detailed below and in Table 22), we manually checked multiple samples at the boundaries of these thresholds. Based on this evaluation, the clips were categorized into three distinct groups and the text-video visualization results satisfying different threshold conditions are shown in Figure 17.

- **High-Quality Clips** ($C_H$)**:** These clips were designated as high-quality and served as candidates for the DPO dataset. To ensure scenario diversity and account for varying annotation precision across different original data sources, clips were selected if their VQAScore Lin et al. (2024) exceeded a specific threshold $\tau_{j,k}$ defined for its original data source $j$ and motion aspect $k$ (e.g., $k \in \{\text{'object', 'camera'}\}$). These VQAScore thresholds $\tau_{j,k}$ are detailed in Table 22. The set $C_H$ is formally defined as:

$$C_H = \{c \in C_A \mid \text{ExceedsVQAScoreThreshold}(c)\} \tag{9}$$

  where ExceedsVQAScoreThreshold($c$) holds if the VQAScore of clip $c$ (from source $j$, for aspect $k$) is greater than the specific threshold $\tau_{j,k}$ for that source-aspect pair, as given in Table 22.

- **Low-Quality Clips** ($C_L$)**:** Clips that failed to meet the minimum quality criteria were eliminated. This includes clips whose VQAScore was below a human-set minimum threshold $\tau_{vL} = 0.3$. The set $C_L$ is defined as:

$$C_L = \{c \in C_A \mid \text{VQAScore}(c) < 0.3\} \tag{10}$$

- **Instruction Dataset Clips** ($C_S$)**:** The remaining clips formed the SFT instruction dataset. This set is defined as:

$$C_S = C_A \setminus (C_H \cup C_L) \tag{11}$$

**SFT dataset construction and question types.**    The SFT dataset is constructed using the Instruction Dataset Clips ($C_S$). For each clip in $C_S$, we generate a question-answer pair to fine-tune the model's ability to understand and describe motion. To cover diverse aspects of motion understanding, we categorize our questions into three types: Object-centric, Camera-centric, and Mixed-focus. During the SFT data generation, one question is randomly selected from the pool of questions corresponding to the primary motion aspect (object, camera, or mixed) identified in the clip's annotation. These SFT dialogues (question-answer pairs) are crucial as they also form the foundation for constructing the preference data for DPO.

OBJECT-CENTRIC QUESTIONS:    These questions focus on the movement, actions, and interactions of objects within the video. Examples include:

> **Object-centric Questions.**
>
> "What objects are moving in this video?"
> "Can you describe the motion of objects in this video?"
> "What is happening to the objects in this scene?"
> "How are the objects moving in this video?"
> "Describe the movements of the main subjects in this clip."
> "What actions are being performed by the objects in this video?"
> "How would you characterize the object motion in this scene?"
> "What kind of movement do you observe from the objects in this video?"
> "Describe the trajectory of the moving objects in this clip."
> "How do the objects interact with each other in this video?"

CAMERA-CENTRIC QUESTIONS:    These questions probe the camera's movement, techniques, and perspective. Examples include:

> **Camera-centric Questions.**
>
> "How is the camera moving in this video?"
> "Describe the camera motion in this video."

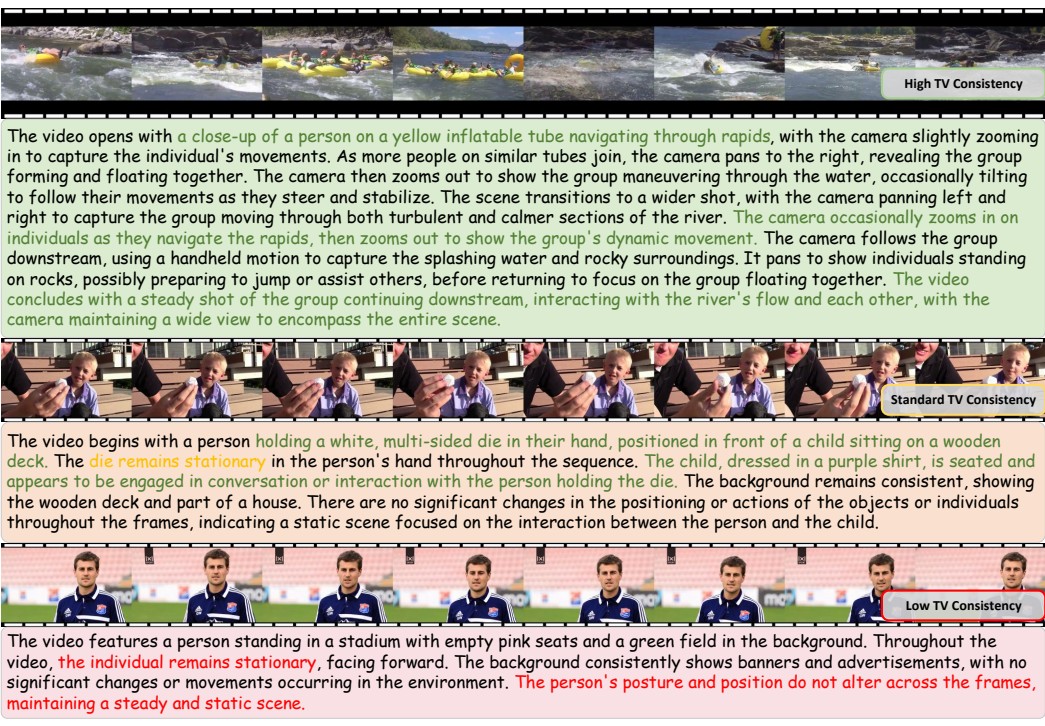

Figure 17: The results corresponding to our three different thresholds are presented separately. **Top:** High consistency between text and video. The camera movement and changes in viewpoint are strictly described in chronological order, resulting in extremely high quality. **Middle:** Fairly good consistency between text and video. The actions of the main characters are described with reasonable accuracy, but some imprecise areas exist. **Bottom:** Relatively poor consistency between text and video, providing limited or erroneous information.

"What camera techniques are used in this video?"
"Is the camera stationary or moving in this clip?"
"How does the camera angle change throughout this video?"
"What kind of camera movements can you identify in this footage?"
"How would you characterize the camera work in this video?"
"Does the camera follow any specific subject in this video?"
"What perspective does the camera provide in this scene?"
"How does the camera movement contribute to the viewing experience?"

MIXED-FOCUS QUESTIONS: These questions require a comprehensive understanding of the interplay between object motion and camera work. Examples include:

Mixed-focus Questions.

"Describe the primary object's specific action, including its fine-grained motion. How does the camera's movement (e.g., tracking, zoom, pan) follow or frame this object's action, and what are the object's key visual attributes highlighted by this interplay?"
"Considering the primary object's movement and its interaction with other elements, what is its implied goal? How does the camera's perspective (e.g., close-up, wide shot, point-of-view) and any dynamic changes in its movement contribute to or obscure this implied intention?"
"Analyze a significant change in the primary object's motion or behavior. How does the camera's operation (e.g., a sudden zoom, a switch to slow motion, a change in focus) coincide with and emphasize this specific change in the object's action?"
"Discuss the overall pattern of the primary object's movement throughout a key segment of the video. Correlate this with the dominant camera movement strategy used in that segment. How does this

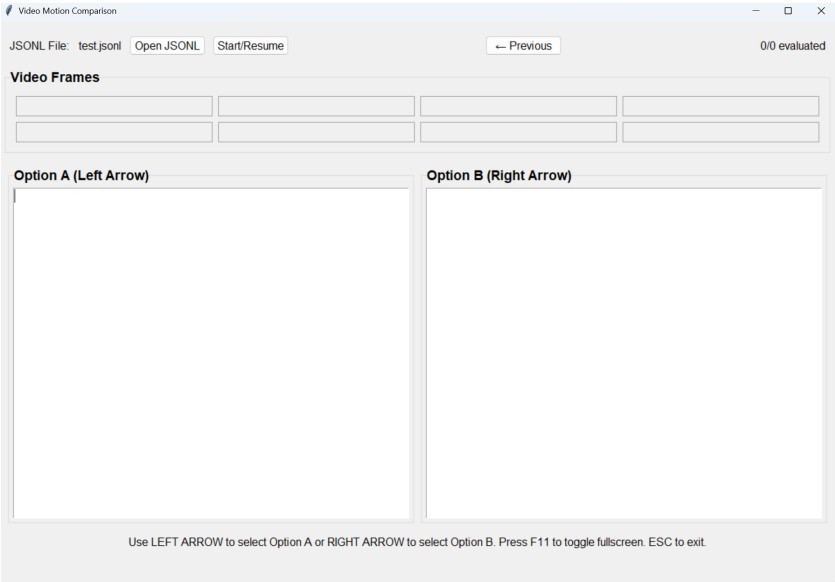

Figure 18: Interactive annotation interface for DPO focused on fine-grained video motion understanding. This Python-based front-end allows annotators to choose between two textual descriptions ("Option A" and "Option B") for the same video clip, selecting the one that more accurately captures the nuanced motion in the video. The interface supports loading data in JSONL format and records annotator preferences, thereby providing data for the model's preference learning.

Table 22: VQAScore Thresholds ($\tau_{j,k}$) for High-Quality Clip Selection ($C_H$), per Source and Motion Aspect. A clip $c$ from source $j$ and aspect $k$ is included in $C_H$ if its VQAScore$(c) > \tau_{j,k}$.

| Data Source ($j$) | Object Motion ($\tau_{j,\text{object}}$) | Camera Motion ($\tau_{j,\text{camera}}$) |
|---|---|---|
| Kinetics-700 | 0.75 | 0.70 |
| ActivityNet | 0.75 | 0.72 |
| Charades | 0.72 | 0.70 |
| Charades-Ego | 0.72 | 0.70 |
| SSV2 | 0.68 | 0.68 |
| OpenVid-1M | 0.70 | 0.70 |

combined object-camera choreography affect the scene's narrative or the information conveyed about the object's activity?"

**Preference dataset construction.** The preference dataset, consists of preference pairs of the form $(x_i, y_i^{chosen}, y_i^{reject})$. These pairs were generated from the high-quality clips in $C_H$. The process involved re-annotating these clips using Tarsier2 Yuan et al. (2025) and then incorporating human preference signals, as illustrated in Figure 5 of the main paper and more details in Section F.

This rigorous, multi-stage curation methodology ensures the high quality of the `MotionVid − QA` subsets, which are crucial for robust model training and evaluation in fine-grained motion understanding.

## F  GUIDELINES FOR HUMAN PREFERENCE ANNOTATION

For DPO, annotators chose between two textual descriptions for a video clip, selecting the one that better captured its fine-grained motion. To ensure fairness, the order in which these two descriptions were presented was randomized. The following guidelines ensured consistent, high-quality annotations: To facilitate this process, we developed an interactive, python-based front-end for

Table 23: Key criteria for human preference annotation in selecting textual descriptions of fine-grained motion. Annotators chose the description that better satisfied these aspects.

| Criterion | Guideline for Selection | Key Questions |
|---|---|---|
| **1. Accuracy** | Prefer more accurate identification & description of primary motion(s). | - Core action correctly identified?
- Agents/objects in motion correct?
- Avoids misinterpreting actions? |
| **2. Granularity** | Prefer more fine-grained & detailed account of motion, capturing nuances. | - Complex movements broken down?
- Specific body/object details?
- Overly general or specific? |
| **3. Temporal Dynamics** | Prefer better capture of temporal aspects (sequence, duration, speed, rhythm). | - Sub-actions order correct?
- Pace/intensity conveyed?
- Speed/tempo changes reflected? |
| **4. Camera Movement** | Prefer description that accurately identifies & describes significant camera movements (e.g., pan, tilt, zoom, tracking). | - Camera movement (pan, tilt, zoom, dolly, static) correctly identified?
- Effect of camera movement on scene understanding clear?
- Distinguished from object motion? |
| **5. Factual Correctness** | Prefer response factually grounded in visual evidence, no hallucinations. | - Only visible elements/actions?
- Contradicts visual information?
- Infers unobservable intent? |

user-friendly annotation, as shown in Figure 18. The relevant code can be found in the supplementary materials.

Annotators selected the preferred response based on a holistic evaluation of the criteria in Table 23, prioritizing superior understanding and articulation of fine-grained motion.

Annotators were advised to review clips multiple times and compare descriptions against these criteria. When responses excelled in different areas, they selected the one most helpful for understanding fine-grained motion.

Furthermore, we randomly selected 100 samples for independent re-annotation by different annotators. The calculated Cohen's $\kappa$ coefficient is 0.731. This demonstrates high consistency in qualitative annotation tasks, and our approach achieves a clearly usable level.

## G  FAILURE CASE

In scenarios where clear visual effects cannot be obtained, such as low-light conditions or extreme weather (e.g., heavy fog or torrential rain), the performance of all vision-perception-based MLLMs will degrade, including current SOTA models and our method.

## H  LIMITATIONS

Given that the accuracy of `MotionSight` is partially dependent on object detection methods, it requires fine-tuning the detection model for domain-specific instances, and its benefits diminish in scenes without objects. Furthermore, we plan to expand our `MotionVid − QA` and improve its quality to train more powerful fine-grained motion understanding models to enhance its impact.

## I  LLMs USAGE

LLMs were used as a general-purpose assist tool to refine the writing of this paper. This usage was strictly limited to improving the clarity, grammar, and overall readability of the text. Specifically, LLMs were employed to rephrase sentences, suggest alternative wording, and correct grammatical errors.

