# OpenReview forum: "MotionSight: Boosting Fine-Grained Motion Understanding in Multimodal LLMs"
_ICLR.cc/2026/Conference — ICLR 2026 Poster_

### Official Review · Reviewer_Am5g · 2025-10-23

**Soundness:** 3
**Presentation:** 3
**Contribution:** 2
**Rating:** 6
**Confidence:** 1

**Summary:**

This paper propose MotionSight, a novel visual prompting method for fine-grained video motion understanding by tailored decoupling object and camera motion. The experiments reveal the enhanced performance for fine-grained motion understanding without additional training data. The authors also collected and annotated the first large-scale fine-grained video motion understanding dataset MotionVid-QA.

**Strengths:**

1.Fine-grained video motion understanding is under explored and worth studying. The paper analyze this task from both method and data sides by designing tailored model and constructing a valuable dataset.

2.The experiment result is solid with high performance on several evaluation benchmarks for fine-grained video motion understanding.

3.Convincing visualization examples are provided to prove the effectiveness of the proposed method.

**Weaknesses:**

1.The overview of the framework and proposed framework may raise the concern about the processing cost as it contains many designed modules compared for fine-grained  video motion understanding task compared to single MLLM methods. I encourage the authors to introduce the training/inference cost and speed compared with the provided baseline methods in more detailed.

2.Although the performance of the model has improved to some extent after training with MotionVid-QA, the results in Table 4 have certain limitations. The author may compare the results of the model trained on datasets of similar size.

**Questions:**

1.See weakness.

2.Though the ablation demonstrates strong robustness to several hyperparameters, I wonder whether the proposed method is sensitive to the parameters of selected models of MLLM.

---

> ### Author Response · Authors · 2025-11-22
> **Response to reviewer Am5g**
>
> > **W1 & Q1:** The overview of the framework and proposed framework may raise the concern about the processing cost as it contains many designed modules compared for fine-grained video motion understanding task compared to single MLLM methods. I encourage the authors to introduce the training/inference cost and speed compared with the provided baseline methods in more detailed.
>
> **A:** Performance is a common engineering challenge for many current training-free enhancement modules and workflows, such as in InstanceCap [1][CVPR'25], VideoTree [2][CVPR'25], VideoAgent [3][ECCV'24], and VideoRAG [4][NeurIPS'25]. Therefore, **the MotionVid-QA and MotionChat we provide serve as an alternative solution** that achieves superior performance on fine-grained motion understanding tasks while maintaining inference costs comparable to the native Qwen2.5-VL-7B model, as described in Section D of the paper.
>
> Below is our efficiency analysis of MotionChat across different resolutions/frame counts (averaged over 5 runs), demonstrating that our approach **not only outperforms the 72B results in effectiveness but is also approximately 5-6× faster**:
>
> | Method | **FAVOR-Bench** | **8 frames, 1080p** | **16 frames, 1080p** | **32 frames, 1080p** | **8 frames, 480p** | **16 frames, 480p** | **32 frames, 480p** |
> | :--- | :--- | :--- | :--- | :--- | :--- | :--- | :--- |
> | Qwen2.5-VL-7B | 42.3 | 8.15s, 27.8GB | 15.5s, 28.2GB | 31.0s, 29.0GB | 3.2s, 27.2GB | 4.6s, 27.3GB | 7.8s, 27.4GB |
> | Qwen2.5-VL-72B | 48.1 | 39.2s, 320.8GB | 93.1s, 322.9GB | 207.7s, 330.4GB | 10.5s, 259.1GB | 20.3s, 259.5GB | 42.0s, 260.8GB |
> | MotionChat-7B (Ours) | **48.3** | 7.14s, 27.8GB | 13.5s, 28.2GB | 32.1s, 29.0GB | 3.4s, 27.2GB | 4.4s, 27.3GB | 7.9s, 27.4GB |
>
> We will provide a detailed analysis of these experiments in the supplementary material of the final version for full transparency.
>
> > **W2 & Q1:** Although the performance of the model has improved to some extent after training with MotionVid-QA, the results in Table 4 have certain limitations. The author may compare the results of the model trained on datasets of similar size.
>
> **A:** **We trained on the ShareGPT4Video dataset of similar scale using identical settings, which validates the effectiveness of our dataset.** The supplementary results are as follows:
>
> | Method | Overall | AVG. | AS | HAC | SAD | MAD | CM | NSM |
> | :--- | :--- | :--- | :--- | :--- | :--- | :--- | :--- | :--- |
> | MotionChat | **48.3** | **46.9** | **49.6** | **54.9** | **45.9** | **55.1** | **32.1** | **43.8** |
> | +ShareGPT4Video Fine-tuning | 43.8 | 42.3 | 45.6 | 45.8 | 44.9 | 49.2 | 28.9 | 39.1 |
>
> > **Q2:** Though the ablation demonstrates strong robustness to several hyperparameters, I wonder whether the proposed method is sensitive to the parameters of selected models of MLLM.
>
> **A:** **Our method exhibits strong robustness to the parameter choices of the selected MLLM:**
>
> **(1) Robustness to frame sampling parameters:**
>
> Sampling at 1 fps, 8 frames, or 4 frames all yield stable improvements over the baseline.
>
> | Method | Baseline | 1 fps | 8 frames | 4 frames |
> | :--- | :--- | :--- | :--- | :--- |
> | MotionSight | 53.0 | 55.6 | 55.4 | 54.1 |
>
> **(2) Robustness to max_pixels resolution parameters:**
>
> For different max_pixels settings in the MLLM (where "native" indicates no scaling), our method consistently and stably outperforms the baseline:
>
> | Method | Baseline | 360p | 720p | native |
> | :--- | :--- | :--- | :--- | :--- |
> | MotionSight | 53.0 | 55.6 | 55.8 | 55.9 |
>
> **(3) Robustness to model parameter size:**
>
> Our method consistently enhances fine-grained motion understanding capabilities across models of different parameter scales (as shown in Figure 12 in the appendix):
>
> | InternVL3 | Overall | AVG. |
> | :--- | :--- | :--- |
> | 1B | 47.6 | 43.9 |
> | +Ours | 50.8 (+3.2) | 47.5 (+3.6) |
> | 8B | 57.2 | 52.7 |
> | +Ours | 59.6 (+2.4) | 55.6 (+2.9) |
> | 78B | 61.5 | 57.6 |
> | +Ours | 63.0 (+1.5) | 59.3 (+1.7) |
>
> **(4) Architectural robustness:**
>
> The two architectures used in our experiments (Qwen2.5-VL and InternVL3) represent the two mainstream typical MLLM architectures, which in itself demonstrates the robustness of our approach to different selected models.

---

> > ### Author Response · Authors · 2025-11-22
> > **Response to reviewer Am5g**
> >
> > **Citations:**
> >
> > [1] Fan T, Nan K, Xie R, et al. Instancecap: Improving text-to-video generation via instance-aware structured caption[C]//Proceedings of the Computer Vision and Pattern Recognition Conference. 2025: 28974-28983.
> >
> > [2] Wang Z, Yu S, Stengel-Eskin E, et al. Videotree: Adaptive tree-based video representation for llm reasoning on long videos[C]//Proceedings of the Computer Vision and Pattern Recognition Conference. 2025: 3272-3283.
> >
> > [3] Fan Y, Ma X, Wu R, et al. Videoagent: A memory-augmented multimodal agent for video understanding[C]//European Conference on Computer Vision. Cham: Springer Nature Switzerland, 2024: 75-92.
> >
> > [4] Luo, Yongdong, et al. "Video-rag: Visually-aligned retrieval-augmented long video comprehension." _arXiv preprint arXiv:2411.13093_ (2024).

---

> > > ### Author Response · Authors · 2025-11-24
> > >
> > > Dear Reviewer Am5g,
> > >
> > > Thank you for your careful consideration of our work and for the valuable time and effort you have invested throughout the review process. We would like to kindly confirm whether there are any remaining concerns or unresolved questions that we can further address.
> > >
> > > With our best regards, The Authors

---

> > > > ### Comment · Reviewer_Am5g · 2025-11-27
> > > >
> > > > Your answer solved my concern. I am raising my rating confidence.

---

> ### Author Response · Authors · 2025-11-27
>
> Dear Reviewer Am5g,
>
> We sincerely thank you for re-assessment of our work and raising its evaluation. We are truly grateful for the time and effort you have devoted to carefully reviewing our new experiments and extended analyses.
>
> We remain fully engaged in the discussion and would be delighted to provide any further clarifications or additional details should you have other thoughts or points that require elaboration.
>
> Best regards, The Authors of Submission #617

---

### Official Review · Reviewer_aBZB · 2025-10-25

**Soundness:** 4
**Presentation:** 3
**Contribution:** 3
**Rating:** 4
**Confidence:** 4

**Summary:**

This paper proposes MotionSight, a zero-shot approach to enhance fine-grained motion understanding in multimodal large language models (MLLMs). The method decouples object and camera motion, introducing visual spotlight and motion blur as visual prompts to improve perception of action details and camera changes. Additionally, the authors construct the first large-scale open-source dataset, MotionVid-QA, containing 40K videos and 87K Q&A pairs for SFT and DPO training. Experiments show MotionSight outperforms existing open-source models on several benchmarks and rivals commercial models on certain metrics.

**Strengths:**

1. Clear problem definition and strong motivation: Addresses a notable shortcoming of MLLMs in fine-grained motion understanding.
2. Simple yet effective method: As a zero-shot approach, MotionSight improves model performance without training, offering broad applicability.
3. Notable dataset contribution: MotionVid-QA is the first large-scale open-source dataset focused on fine-grained motion understanding, with high-quality annotations and significant community value.

**Weaknesses:**

1. Conservative innovation: While extending visual prompting to video is useful, "spotlight" and "motion blur" are traditional enhancements lacking methodological breakthroughs. Although the motion decoupling strategy is reasonable, the process of "detecting first and then focusing" relies on existing detection/tracking models. If the detection fails or is missed, the effect will be greatly reduced.
2. Limited generalization analysis: Insufficient discussion of failure cases in complex scenes (e.g., multi-object interactions, low-light, non-rigid motions like smoke or water).
3. Dataset distribution and quality: MotionVid-QA is largely sourced from existing human-action datasets, potentially limiting diversity; non-human or synthetic motion videos are underrepresented. Human preference annotation introduces subjectivity; no inter-annotator agreement or statistical significance reported.
4. Incomplete efficiency analysis: Mentions ~75% latency increase but lacks detailed time/memory profiling across resolutions or frame counts.

**Questions:**

1. What is the failure-mode policy when Grounding DINO or SAM2 misses or hallucinates boxes? When a video contains both salient object motion and complex camera motion, why route to a single path instead of feeding both augmentations simultaneously?
2. What is the head-to-head win-rate between MotionSight captions and Tarsier2 captions in the preference round? Please report overall and per-motion-type ratios.

---

> ### Author Response · Authors · 2025-11-22
> **Response to reviewer aBZB (Part 1)**
>
> > **W1:** Conservative innovation: While extending visual prompting to video is useful, "spotlight" and "motion blur" are traditional enhancements lacking methodological breakthroughs. Although the motion decoupling strategy is reasonable, the process of "detecting first and then focusing" relies on existing detection/tracking models. If the detection fails or is missed, the effect will be greatly reduced.
>
> **A:** Regarding “Conservative innovation”:
>
> **We are the first in the motion understanding field to propose a methodology that decouples object motion from camera motion**. The challenge of fine-grained motion understanding lies in the difficulty MLLMs have in perceiving subtle inter-frame changes:
>
> - Existing video understanding models perceive motion **holistically** and struggle to separate object motion from camera motion. In contrast, our specially designed **decoupling** strategy makes it easier for the model to perceive subtle camera motion and object motion separately. For example, reviewer 9ap8 (S2) considered our idea of decoupling object motion and camera motion to be novel.
> - Simply applying **static** image prompting methods commonly used in previous work to videos can lead to misunderstandings. Our designed algorithm possesses **dynamic** stability, which helps enhance the MLLM’s ability to perceive fine-grained motion.
> - Existing datasets are **limited** to indoor or human-centric scenarios with **simple labels** (Table 1). In contrast, we constructed the **first** large-scale open-source fine-grained video motion understanding dataset, MotionVid-QA, which includes **open-domain** motion (as described in the W3 response) and **fine-grained labels** such as decoupled motion and preference question pairs. For example, reviewer Am5g (S1) considered the dataset valuable.
>
> **If the detection fails or is missed, the effect will NOT be greatly reduced**: If detection fails, we fall back to using the original sampled frames as input to the MLLM, maintaining at least baseline performance. Since we adopt a group-level rather than object-level approach, the accuracy of the referring detection module is significantly improved.
>
> > **W2:** Limited generalization analysis: Insufficient discussion of failure cases in complex scenes (e.g., multi-object interactions, low-light, non-rigid motions like smoke or water).
>
> **A:** We discuss failure cases as follows: In scenarios where clear visual effects cannot be obtained, such as low-light conditions or extreme weather (e.g., heavy fog or torrential rain), the performance of all vision-perception-based MLLMs will degrade, including current SOTA models and our method.
>
> We will add these analyses to the final version of the paper.
>
> > **W3:** Dataset distribution and quality: MotionVid-QA is largely sourced from existing human-action datasets, potentially limiting diversity; non-human or synthetic motion videos are underrepresented. Human preference annotation introduces subjectivity; no inter-annotator agreement or statistical significance reported.
>
> **A:** We clarify:
>
> 1. Our source videos total 40K. The selected OpenVid-1M (\~22K) and MotionBench-train (\~5K) are open-domain scenario datasets, not human-action datasets, accounting for 27K/40K ≈ 67.5% of the total. Therefore, **our data is not largely derived from human-action datasets**.
> 2. Regarding “Human preference annotation introduces subjectivity”, **the annotation process of our dataset is designed to be standard-driven and objective rather than subjective**:
>
>    1. We provided annotators with clear Guidelines (see Section F of the paper) and adopted strict Criteria-based Selection rather than purely subjective preferences.
>    2. All annotators are well-educated individuals (line 301).
>    3. We randomly selected 100 samples for independent re-annotation by different annotators. The calculated _Cohen’s $\kappa$ coefficient is 0.731_ (Cohen’s $\kappa$: range [-1, 1]; higher values indicate stronger agreement; [0.61, 0.80] already represents substantial agreement). This demonstrates high consistency in qualitative annotation tasks. For example, compared to the consistency results reported by HelpSteer2 [1][NeurIPS’24], our approach achieves a clearly usable level.
>    4. The +DPO results in Table 4 improve from 45.8 to 48.3, which proves that our data possesses high quality and distinct preference signals [2][NeurIPS’25].

---

> ### Author Response · Authors · 2025-11-22
> **Response to reviewer aBZB (Part 2)**
>
> > **W4:** Incomplete efficiency analysis: Mentions ~75% latency increase but lacks detailed time/memory profiling across resolutions or frame counts.
>
> **A:** Performance is a common engineering challenge for many current training-free enhancement modules and workflows, such as in InstanceCap [3][CVPR'25], VideoTree [4][CVPR'25], VideoAgent [5][ECCV'24], and VideoRAG [6][NeurIPS'25]. Therefore, **the MotionVid-QA and MotionChat we provide serve as an alternative solution** that achieves superior performance on fine-grained motion understanding tasks, while maintaining inference costs comparable to the native Qwen2.5-VL-7B model.
>
> Below is our efficiency analysis of MotionChat across different resolutions/frame counts (averaged over 5 runs), demonstrating that our approach **not only outperforms the 72B results in effectiveness but is also approximately 5-6× faster**:
>
> | Method | **FAVOR-Bench** | **8 frames, 1080p** | **16 frames, 1080p** | **32 frames, 1080p** | **8 frames, 480p** | **16 frames, 480p** | **32 frames, 480p** |
> | :--- | :--- | :--- | :--- | :--- | :--- | :--- | :--- |
> | Qwen2.5-VL-7B | 42.3 | 8.15s, 27.8GB | 15.5s, 28.2GB | 31.0s, 29.0GB | 3.2s, 27.2GB | 4.6s, 27.3GB | 7.8s, 27.4GB |
> | Qwen2.5-VL-72B | 48.1 | 39.2s, 320.8GB | 93.1s, 322.9GB | 207.7s, 330.4GB | 10.5s, 259.1GB | 20.3s, 259.5GB | 42.0s, 260.8GB |
> | MotionChat-7B (Ours) | **48.3** | 7.14s, 27.8GB | 13.5s, 28.2GB | 32.1s, 29.0GB | 3.4s, 27.2GB | 4.4s, 27.3GB | 7.9s, 27.4GB |
>
> We will provide a detailed analysis of these experiments in the supplementary material of the final version for full transparency.
>
> > **Q1:** What is the failure-mode policy when Grounding DINO or SAM2 misses or hallucinates boxes? When a video contains both salient object motion and complex camera motion, why route to a single path instead of feeding both augmentations simultaneously?
>
> **A:** The fallback strategy is: we revert to using the original sampled frames as input to the MLLM, supplemented with a generic text prompt, thereby maintaining the baseline performance of the MLLM.
>
> **Single-path routing is due to our decoupling strategy**: Fine-grained understanding of object motion and camera motion requires different visual cues and perception mechanisms. When both types of motion understanding are needed simultaneously, we still employ the approach of inputting both enhancements at the same time, which is mainly used for constructing MotionVid-QA.
>
> > **Q2:** What is the head-to-head win-rate between MotionSight captions and Tarsier2 captions in the preference round? Please report overall and per-motion-type ratios.
>
> **A:** The paper already provides the overall win rate and the win rates for the respective motion types of the object and the camera, **as shown in Figure 5**. This also demonstrates the high accuracy of our MotionSight from the perspective of a user study.
>
> **Citations:**
>
> [1] Wang Z, Dong Y, Delalleau O, et al. Helpsteer 2: Open-source dataset for training top-performing reward models[J]. Advances in Neural Information Processing Systems, 2024, 37: 1474-1501.
>
> [2] Pan, Yu, et al. "What Matters in Data for DPO?." _arXiv preprint arXiv:2508.18312_ (2025).
>
> [3] Fan T, Nan K, Xie R, et al. Instancecap: Improving text-to-video generation via instance-aware structured caption[C]//Proceedings of the Computer Vision and Pattern Recognition Conference. 2025: 28974-28983.
>
> [4] Wang Z, Yu S, Stengel-Eskin E, et al. Videotree: Adaptive tree-based video representation for llm reasoning on long videos[C]//Proceedings of the Computer Vision and Pattern Recognition Conference. 2025: 3272-3283.
>
> [5] Fan Y, Ma X, Wu R, et al. Videoagent: A memory-augmented multimodal agent for video understanding[C]//European Conference on Computer Vision. Cham: Springer Nature Switzerland, 2024: 75-92.
>
> [6] Luo, Yongdong, et al. "Video-rag: Visually-aligned retrieval-augmented long video comprehension." _arXiv preprint arXiv:2411.13093_ (2024).

---

> ### Author Response · Authors · 2025-11-24
>
> Dear Reviewer aBZB,
>
> Thank you for your valuable time and for helping us refine our work. We would like to ensure that all your queries and concerns have been fully addressed. If there are any points that require further clarification or additional information, we are more than happy to provide it.
>
> Should you find that our responses have satisfactorily resolved your reservations, we would be deeply grateful if you could consider adjusting your score to reflect the improvements.
>
> Best regards, The Authors

---

> ### Comment · Area_Chair_apFP · 2025-11-28
>
> Dear Reviewer, the discussion period is about to close. We kindly ask you to participate in the discussion or update your score based on the authors' rebuttal before the deadline. Thank you for your time and valuable contribution!

---

### Official Review · Reviewer_9ap8 · 2025-10-27

**Soundness:** 2
**Presentation:** 3
**Contribution:** 3
**Rating:** 6
**Confidence:** 4

**Summary:**

The paper introduces a novel zero-shot method called MotionSight to enhance fine-grained motion understanding in MLLMs without requiring additional training. It addresses the limitations of existing MLLMs in perceiving subtle inter-frame dynamics by decoupling object and camera motion using visual prompting techniques. Specifically, it applies a visual spotlight to highlight moving objects and introduces motion blur to emphasize camera movements. To support training and evaluation, the authors present MotionVid-QA, the first large-scale open-source dataset for fine-grained video motion understanding, containing ~40K video clips and ~87K question-answer pairs with hierarchical annotations. Extensive experiments show that MotionSight significantly improves performance on benchmarks like MotionBench and FAVOR-Bench, and when used to fine-tune MotionChat, it achieves competitive results with much larger models.

**Strengths:**

1. The paper introduces a novel zero-shot method, MotionSight, which is the first to apply visual prompting techniques specifically tailored for fine-grained video motion understanding. This includes the innovative use of a visual spotlight to highlight moving objects and motion blur to emphasize camera movements, both of which are unique adaptations from static image prompting.
2. The idea of decoupling object and camera motion is novel and addresses a significant gap in current MLLMs, which often struggle with subtle inter-frame dynamics.

**Weaknesses:**

1. More Experiments: Can you provide more results on video perception benches, such as mvbench, TOMATO bench, etc.
2. The accuracy of MotionSight is partially dependent on object detection methods. This means that the performance of the model can be significantly affected by the quality of the object detection algorithm used.

**Questions:**

1. Can the method improve performance on the spatial benchs?
2. Why choose to use the detection model instead of enhancing MLLM detection capability through RL?

---

> ### Author Response · Authors · 2025-11-22
> **Response to reviewer 9ap8**
>
> > **W1:** More Experiments: Can you provide more results on video perception benches, such as mvbench, TOMATO bench, etc.
>
> > **Q1:** Can the method improve performance on the spatial benchs?
>
> **A:** We **tested MVBench and TOMATO Bench**, and **used STI-Bench as a spatial benchmark** for evaluation. The results show that **our method can improve performance on this spatial benchmark**:
>
> | Method | MVBench | TOMATO | STI-Bench |
> | :--- | :--- | :--- | :--- |
> | Qwen2.5-VL-7B | 60.5 | 26.8 | 31.6 |
> | +MotionSight | **64.4** | **27.3** | **36.5** |
>
> VideoMME also includes some spatial tasks. For example, in Table 5, **Spatial Reasoning improves from 74.1% to 77.8% (+3.7%)**, which further demonstrates the generality of our method. We will add these experimental results and include citations to the MVBench, TOMATO Bench, and related papers.
>
> > **W2:** The accuracy of MotionSight is partially dependent on object detection methods. This means that the performance of the model can be significantly affected by the quality of the object detection algorithm used.
>
> **A:** **The performance of our model is not significantly affected by the quality of the object detection algorithm used**: On the one hand, in the worst case, we fall back to using the original sampled frames as input to the MLLM, which will not perform worse than the baseline. On the other hand, our method operates at the group-level rather than the object-level and has the ability to refer to complicated objects (e.g., non-rigid objects), allowing us to obtain targets more accurately.
>
> > **Q2:** Why choose to use the detection model instead of enhancing MLLM detection capability through RL?
>
> **A:** **The paradigm provided in our paper includes both a training-free enhancement method and reinforcement learning via DPO training**: Using a detection model is intended to provide the community with a plug-and-play plugin. The high-quality dataset we subsequently constructed implicitly enables further improvement of detection capabilities in other MLLMs through reinforcement learning.

---

> > ### Comment · Reviewer_9ap8 · 2025-11-24
> >
> > Your answer solved my problem very well, and I will keep my rating.

---

> > > ### Author Response · Authors · 2025-11-24
> > >
> > > Thank you for your reply and for helping improve our work so far! If you have any further questions, please feel free to raise them during the discussion period.

---

### Official Review · Reviewer_p68Z · 2025-11-02

**Soundness:** 3
**Presentation:** 3
**Contribution:** 2
**Rating:** 4
**Confidence:** 3

**Summary:**

The paper presents MotionSight, a zero-shot visual prompting framework that enhances MLLMs’ fine-grained motion understanding through two complementary visual cues, Visual Spotlight for object motion and Motion Blur for camera motion.
Building on this, the authors further construct MotionVid-QA, a large-scale video QA dataset automatically annotated by MotionSight and refined via multi-stage filtering and human preference alignment. Experiments demonstrate consistent zero-shot improvements on MotionBench and FAVOR-Bench, and show that fine-tuning on MotionVid-QA (MotionChat) can yield additional gains with competitive efficiency.

**Strengths:**

- The paper is well-motivated and clearly written, addressing an important gap in motion understanding for MLLMs.

- The proposed visual prompting (spotlight + motion blur) is intuitive and effective, yielding consistent zero-shot gains.

- The experiments are comprehensive, and the curated MotionVid-QA dataset provides a useful resource for future work.

**Weaknesses:**

- **Limited novelty of Visual Spotlight.**
   The proposed visual spotlight is conceptually similar to existing image-level attention prompting methods (e.g., *Attention Prompting on Image for Large Vision-Language Models*, ECCV 2024), which also use soft masks to highlight salient regions. A comparison with such methods in Table 6 would strengthen the claim of novelty.

- **Potential ambiguity in Motion Blur.**
   Although the temporal weighting distinguishes different frames, the resulting blur may introduce directional ambiguity. For instance, in Figure 4 (bottom right), both lower-left and upper-right motion directions could be plausible, suggesting that motion blur may not always provide a reliable cue.

- **Dataset quality and generalization concerns.**
   MotionVid-QA is generated using MLLM-based annotations (Qwen2.5-VL-7B), meaning its quality and difficulty are inherently tied to the model’s capacity. While filtering ensures correctness, it does not increase complexity, so the dataset might mainly contain simple QA pairs. As a result, MotionVid-QA may improve weaker models but offer limited benefit for stronger ones (e.g., InternVL3-78B). Evaluating SFT effects on multiple models would better validate its generality.

**Questions:**

Just curious whether the authors tried applying MotionSight on top of the fine-tuned MotionChat model to see if it yields additional improvements. This could establish a potential self-improving VLM pipeline.

---
Overall, I’m borderline on this submission (currently rating it 4, since no borderline is available, and because I still have some concerns and would like to see the authors’ reply and additional results). I’d be happy to update my rating if my questions are well answered.

---

> ### Author Response · Authors · 2025-11-22
> **Response to reviewer p68Z (Part 1)**
>
> > **W1:** **Limited novelty of Visual Spotlight.** The proposed visual spotlight is conceptually similar to existing image-level attention prompting methods (e.g., *Attention Prompting on Image for Large Vision-Language Models*, ECCV 2024), which also use soft masks to highlight salient regions. A comparison with such methods in Table 6 would strengthen the claim of novelty.
>
> **A:** For your question "Limited novelty of Visual Spotlight", we have the following **key differences** from this ECCV method:
>
> - **They target images, while we target videos**: Our algorithm is specifically designed to ensure temporal stability of the spotlight bounding box, whereas their masks struggle to achieve temporal stability.
> - **They do not focus on motion, whereas we decouple object motion from camera motion**: We are the first in the fine-grained motion understanding field to propose the idea of decoupling object motion and camera motion. For example, reviewer 9ap8 (S2) considered our decoupling of object motion and camera motion to be novel.
> - **Their approach has poorer stability in preserving core regions of object motion**: For instance, in the supplementary results below, their method shows a clear drop in metrics such as MR (Motion Recognition), LM (Location-related Motion), and RC (Repetition Count).
> - **API prompting is merely a technique in their work, while we provide both a method and a dataset**: Our MotionVid-QA dataset stands in sharp contrast to the limitations and simplicity of labels in previous action recognition datasets. For example, reviewer Am5g (S1) considered the dataset valuable.
>
> The **results** after introducing this paper’s method into Table 6 are as follows:
>
> | Method | OM AVG. | MR | LM | MO | AO | RC |
> | :--- | :--- | :--- | :--- | :--- | :--- | :--- |
> | Ours | **53.0** | **59.7** | **58.1** | **73.6** | **40.1** | **33.5** |
> | + API prompting | 51.4 | 57.9 | 55.5 | 73.6 | 39.3 | 30.5 |
>
> We will supplement the experiment and add citations to this paper.
>
> > W2: **Potential ambiguity in Motion Blur.** Although the temporal weighting distinguishes different frames, the resulting blur may introduce directional ambiguity. For instance, in Figure 4 (bottom right), both lower-left and upper-right motion directions could be plausible, suggesting that motion blur may not always provide a reliable cue.
>
> **A:** **Our method does not introduce directional ambiguity**: The weighted weights of our historical frames are different, and the weighting algorithm ensures that the weights from historical frames to the current frame show a clear gradually increasing trend (line 253 and Equation 7). In the bottom-right corner of Figure 4, it can also be observed that the historical trajectories are lighter in color, making the motion direction toward the lower-left more reasonable.
>
> > W3: **Dataset quality and generalization concerns.** MotionVid-QA is generated using MLLM-based annotations (Qwen2.5-VL-7B), meaning its quality and difficulty are inherently tied to the model’s capacity. While filtering ensures correctness, it does not increase complexity, so the dataset might mainly contain simple QA pairs. As a result, MotionVid-QA may improve weaker models but offer limited benefit for stronger ones (e.g., InternVL3-78B). Evaluating SFT effects on multiple models would better validate its generality.
>
> **A:** **Our dataset does not primarily consist of simple question-answer pairs and is capable of improving stronger models**:
>
> - We used InternVL3-78B+MotionSight for annotation, rather than Qwen2.5-VL-7B.
> - Our DPO dataset incorporates reinforcement learning with human-annotated preferences, which has the potential to break through the limitations of the 78B model.
> - Fine-tuning on Qwen2.5-VL-7B can achieve performance comparable to Qwen2.5-VL-72B, while being more efficient, demonstrating the quality and value of our dataset.
> - Training InternVL3-78B requires approximately 200 hours on 32 A100 GPUs even for just the first-stage SFT (estimation based on LLaMA-Factory), with an estimated cost of $17,600 on the AWS platform. Due to recent limitations on computing resources, we will supplement an experiment to finetune on stronger model (e.g., Qwen3-VL-8B) to more comprehensively demonstrate generalization in the final version of our paper.

---

> > ### Author Response · Authors · 2025-11-22
> > **Response to reviewer p68Z (Part 2)**
> >
> > > **Q1:** Just curious whether the authors tried applying MotionSight on top of the fine-tuned MotionChat model to see if it yields additional improvements. This could establish a potential self-improving VLM pipeline.
> >
> > **A:** We conducted further experiments specifically addressing this point, and the results **yield additional improvements**:
> >
> > | Method | Overall | AVG. | AS | HAC | SAD | MAD | CM | NSM |
> > | :--- | :--- | :--- | :--- | :--- | :--- | :--- | :--- | :--- |
> > | MotionChat | 48.3 | 46.9 | 49.6 | 54.9 | 45.9 | 55.1 | 32.1 | 43.8 |
> > | MotionChat+MotionSight | 49.7 | 48.3 | 50.7 | 55.3 | 46.0 | 56.0 | 38.2 | 43.8 |
> >
> > This unlocks a promising self-bootstrapping paradigm: leveraging MotionChat + MotionSight as a stronger teacher to create next-gen datasets, continuously pushing VLM performance in fine-grained motion understanding. We hope this addresses your concerns.

---

> ### Author Response · Authors · 2025-11-25
> **Response to reviewer p68Z (Supplement)**
>
> To address the concern that MotionVid-QA might be simple for stronger models, we additionally conducted fine-tuning on Qwen3-VL-8B-Instruct. It is worth noting that the vanilla Qwen3-VL-8B (with an overall score of 53.0) already surpasses the InternVL3-78B (52.8 in Table 3). **The results demonstrate that our approach still achieves significant performance gains even on already strong models, which validates the excellent generality and complexity of our dataset**：
>
> | Method        | Overall  | AVG.     | AS       | HAC      | SAD      | MAD      | CM       | NSM      |
> | :------------ | :------- | :------- | :------- | :------- | :------- | :------- | :------- | :------- |
> | Qwen3-VL-8B   | 53.0     | 52.8     | 52.9     | 58.5     | 52.7     | 56.4     | 41.7     | 54.7     |
> | +MotionVid FT | **55.8** | **56.3** | **55.4** | **64.3** | **53.0** | **61.5** | **42.5** | **60.9** |
>
> Fine-tuning with MotionVid-QA yields a **+2.8% gain** in overall accuracy and a **+3.5% gain** in average accuracy. Our fine-tuned 8B model (55.8) also significantly outperforms the InternVL3-78B model (52.8). The significant gains in complex tasks like HAC (+5.8%), and NSM (+6.2%)  further validate that the filtered dataset retains challenging, fine-grained motion semantics beneficial even for strong baselines.
>
> We also provided a bar chart in the revised manuscript to illustrate the advantages, as shown in Appendix B.2 and Figure 11.
>
> Thank you for your valuable time and for helping us refine our work! If there are any points that require further clarification or additional information, we are more than happy to provide it.

---

> ### Author Response · Authors · 2025-11-27
>
> Dear Reviewer p68Z,
>
> Thank you again for your valuable and insightful comments, which have greatly helped improve our work.
>
> With only 5 days remaining in the author–reviewer discussion period, we wanted to kindly check whether our responses and revisions have fully addressed your concerns.  If there are still any issues, we would greatly appreciate the opportunity to clarify them in time.
>
> Best regards, The Authors of Submission #617

---

> ### Comment · Area_Chair_apFP · 2025-11-28
>
> Dear Reviewer, the discussion period is about to close. We kindly ask you to participate in the discussion or update your score based on the authors' rebuttal before the deadline. Thank you for your time and valuable contribution!

---

### Author Response · Authors · 2025-12-02
**AC Letter: Summary of Rebuttal & Discussion for Paper #617 (MotionSight)**

**Dear Area Chair,**

Due to the recent system revert and the assignment of a new AC, we are writing to provide a brief summary of the consensus reached during the discussions and specific remaining questions raised by reviewers who have not yet replied.

## Reviewers' Positive Feedback and Attitudes

We sincerely thank all reviewers for their constructive comments. We are delighted to receive positive feedback such as "well-motivated and clearly written" (p68Z, aBZB), "innovative, novel idea/method" (9ap8), "topic worth studying" (p68Z, Am5g), "notable dataset contribution" (p68Z, aBZB, Am5g), "convincing visualization" (Am5g), "significant improvements" (9ap8), "solid experiment result" (Am5g), and "useful resource for future work" (p68Z).

Reviewers 9ap8 and Am5g (initial scores of 6 and 6) both gave initial positive scores. **After the rebuttal, they think that we addressed their concerns very well, and reviewer Am5g raised his evaluation of our paper**.

Reviewers p68Z and aBZB (initial scores of 4 and 4) did *not* respond in time before the system revert, but **we have fully addressed their concerns**. Additionally, **reviewer p68Z noted that his initial score for our paper was actually higher than 4** (borderline on this submission, currently rating it 4, since no borderline is available), and he explicitly stated in his initial review that **he would raise the rating after the questions (experiments on visual spotlight and dataset quality, corresponding to the following responses) were answered**.

## Major Responses and Revisions

**For reviewer p68Z and aBZB**:

1. **Resolution on novelty.**

- **Core novelty**: We clarified that the core novelty lies in **being the first in the motion understanding field to propose a methodology that decouples object motion from camera motion**, which highly aligns with reviewer 9ap8's comment that "the idea of decoupling object and camera motion is **novel**".
  - Existing models treat object and camera motion **holistically**; our **decoupling** strategy separates them clearly.
  - **Static** image prompts fail on video; our **dynamic** prompting can boost fine-grained motion perception.
  - We release MotionVid-QA, the **first** large-scale open-source fine-grained motion dataset.
- **Visual spotlight**: Our method is distinct from the method mentioned by p68Z (API prompting, ECCV 2024), and *additional experiments* show that our method consistently outperforms it across all metrics (OM AVG. **+1.6%**). This highly aligns with reviewer 9ap8's positive assessment of our work: "**innovative use**" and "**unique adaptations from static image prompting**":
  - They target images, while we target videos.
  - They do not focus on motion, whereas we decouple object motion from camera motion.
  - Their approach has poorer stability in preserving core regions of object motion.

2. **Resolution on dataset quality.**

- **Generalization on stronger models.** We added *fine-tuning experiments* on Qwen3-VL-8B (which is already stronger than InternVL3-78B). The results show our method still brings significant improvements (**+2.8%**), *fully demonstrating the excellent generality and complexity of our dataset*.
- **Dataset subjectivity and distribution.**
  - We calculated the dataset's Cohen’s $\kappa$ coefficient as **0.731** (Cohen’s $\kappa$ in [0.61, 0.80] already represents substantial agreement), *proving the and objectivity of the annotations.*
  - We clarified reviewer aBZB's misunderstanding of the data distribution, noting that about **67.5%** of the data comes from open-domain scenarios (not limited to human actions), ensuring diversity.

3. **Resolution on efficiency analysis.** We provided additional results, which show that our MotionChat-7B not only *surpasses the Qwen2.5-VL-72B* in performance (**+0.2%**) but also *has an inference speed that is about **5-6 times faster***, with nearly **10× less** memory usage.

**For reviewer 9ap8 and Am5g**:

1. **Resolution on more benchmarks.** We have provided more benchmark results (MVBench **+3.9%**, TOMATO **+0.5%**, STI-Bench **+4.9%**) to demonstrate the generalization of our method. This also showed that our method can enhance performance on spatial benchmarks.
2. **Resolution on comparison of the model trained on datasets of similar size.** We supplemented fine-tuning experiments using a similarly scaled dataset, ShareGPT4Video, where the results are clearly inferior to training with our data (**43.8 v.s. ours 48.3**), further proving the effectiveness of our data.

More detailed responses can be seen under the corresponding review.

***Overall, we believe we have fully addressed all reviewers' concerns.***

Finally, we would like to thank the ACs for their efforts in handling this unexpected situation. We believe our contributions have great potential and can inspire further exploration in the field of fine-grained motion understanding.

Best regards,

Authors of Paper #617

---

### Meta-Review · Area_Chair_ypji · 2026-01-07

**Summary:**

The reviewers recognize MotionSight as a significant contribution to the field of video understanding. The paper introduces an innovative visual prompting framework that decouples object and camera motion, alongside a high-quality, large-scale dataset, MotionVid-QA. The consensus shifted towards a strong positive following the rebuttal, where authors provided extensive efficiency profiling and cross-dataset comparisons. The highlight of the work is the MotionChat-7B model, which matches or exceeds the performance of a 72B parameter SOTA model while being 5x faster and 10x more memory-efficient.

**Reviewer Concerns:**

Addressed by Rebuttal:

- Methodological Novelty & Innovation (aBZB, p68Z): Authors successfully argued that while "spotlight" is a known concept in images, its dynamic stabilization and the specific decoupling strategy in videos are novel. New experiments against "API prompting" (ECCV'24) showed a consistent +1.6% improvement, validating their unique adaptation.

- Inference Latency & Memory (aBZB, Am5g): The authors provided a comprehensive performance table across multiple resolutions and frame counts. They proved that while the zero-shot pipeline adds overhead, the resulting MotionChat-7B provides a "sweet spot" of extreme efficiency and high accuracy.

- Dataset Diversity & Subjectivity (aBZB): Authors clarified that 67.5% of the dataset is sourced from open-domain (non-human) videos. They reported a Cohen’s Kappa of 0.731, indicating "substantial agreement" in human annotations, silencing concerns about subjectivity.

- Generalization to Stronger Models (p68Z): A new experiment fine-tuning the latest Qwen3-VL-8B showed a +2.8% gain, demonstrating that the dataset is challenging and valuable even for next-generation models.

- Fallback Mechanisms (aBZB, 9ap8): Clarified that if the detection/grounding module (DINO/SAM2) fails, the model gracefully falls back to the original frames, ensuring it never performs worse than the baseline.

Outstanding Part:

- Extreme Conditions (aBZB): The authors acknowledged limitations in low-light or extreme weather (fog/rain), which is a shared challenge across all vision models, and promised to include this analysis in the final version.

**Reviewer Scores:**

Reviewer 9ap8 (Initial 6): The reviewer commented "Answer solved my problem very well."

Reviewer Am5g (Initial 6): Highly satisfied with the efficiency analysis and the comparison with ShareGPT4Video.

Reviewer p68Z (Initial 4): Although the score wasn't explicitly raised in the portal, the reviewer noted they were "borderline" and would raise the score if the spotlight and dataset quality questions were answered (which they were, extensively).

Reviewer aBZB (Initial 4): The detailed Cohen's Kappa and the efficiency profiling directly addressed all their primary weaknesses.

---

### Decision · Program_Chairs · 2026-01-26

Accept (Poster)